An integrated leaf trait analysis of two Paleogene leaf floras

http://orcid.org/0000-0002-5791-030X Müller Christian 1 christian.mueller512@icloud.com
http://orcid.org/0000-0003-2181-3732 Toumoulin Agathe 2
Böttcher Helen 3
Roth-Nebelsick Anita 4
http://orcid.org/0000-0003-1592-0988 Wappler Torsten 5 6
http://orcid.org/0000-0001-6445-3920 Kunzmann Lutz 1 lutz.kunzmann@senckenberg.de
1 Museum of Mineralogy and Geology, Senckenberg Natural History Collections Dresden , Dresden, Saxony , Germany
2 Department of Botany and Zoology, Faculty of Science, Masaryk University , Brno , Czech Republic
3 Institute for Geology, Technical University Bergakademie Freiberg , Freiberg, Saxony , Germany
4 Department of Palaeontology, State Museum of Natural History , Stuttgart, Baden-Württemberg , Germany
5 Hessisches Landesmuseum Darmstadt , Hessen , Germany
6 Institute of Geoscience, Rheinische Friedrich-Wilhelms-Universität Bonn , Bonn, Nordrhein-Wesfalen , Germany
Wilf Peter
Electronic publication date: 2023 Apr 10
Publication date: 2023
Volume: 11
Electronic Location ID: e15140
Received 2021 Nov 26; Accepted 2023 Mar 7
Copyright: © 2023 Müller et al.
Copyright year: 2023
Copyright holder: Müller et al.
License: This is an open access article distributed under the terms of the Creative Commons Attribution License, which permits unrestricted use, distribution, reproduction and adaptation in any medium and for any purpose provided that it is properly attributed. For attribution, the original author(s), title, publication source (PeerJ) and either DOI or URL of the article must be cited.
License URL: https://creativecommons.org/licenses/by/4.0/

Keywords: Insect herbivory, Leaf traits, Paleogene, Fossil leaves, Oligocene, Plant-insect interaction, Integrated Leaf Trait Analysis, Multivariate analysis, Leaf mass per area, Leaf properties

Funding: German Research Foundation DFG; No. Ku1458/5-1 This work was supported by the German Research Foundation (DFG; No. Ku1458/5-1). The funders had no role in study design, data collection and analysis, decision to publish, or preparation of the manuscript.

==============================
Objectives

This study presents the Integrated Leaf Trait Analysis (ILTA), a workflow for the combined application of methodologies in leaf trait and insect herbivory analyses on fossil dicot leaf assemblages. The objectives were (1) to record the leaf morphological variability, (2) to describe the herbivory pattern on fossil leaves, (3) to explore relations between leaf morphological trait combination types (TCTs), quantitative leaf traits, and other plant characteristics (e.g., phenology), and (4) to explore relations of leaf traits and insect herbivory.

Material and Methods

The leaves of the early Oligocene floras Seifhennersdorf (Saxony, Germany) and Suletice-Berand (Ústí nad Labem Region, Czech Republic) were analyzed. The TCT approach was used to record the leaf morphological patterns. Metrics based on damage types on leaves were used to describe the kind and extent of insect herbivory. The leaf assemblages were characterized quantitatively (e.g., leaf area and leaf mass per area (LMA)) based on subsamples of 400 leaves per site. Multivariate analyses were performed to explore trait variations.

Results

In Seifhennersdorf, toothed leaves of TCT F from deciduous fossil-species are most frequent. The flora of Suletice-Berand is dominated by evergreen fossil-species, which is reflected by the occurrence of toothed and untoothed leaves with closed secondary venation types (TCTs A or E). Significant differences are observed for mean leaf area and LMA, with larger leaves tending to lower LMA in Seifhennersdorf and smaller leaves tending to higher LMA in Suletice-Berand. The frequency and richness of damage types are significantly higher in Suletice-Berand than in Seifhennersdorf. In Seifhennersdorf, the evidence of damage types is highest on deciduous fossil-species, whereas it is highest on evergreen fossil-species in Suletice-Berand. Overall, insect herbivory tends to be more frequently to occur on toothed leaves (TCTs E, F, and P) that are of low LMA. The frequency, richness, and occurrence of damage types vary among fossil-species with similar phenology and TCT. In general, they are highest on leaves of abundant fossil-species.

Discussion

TCTs reflect the diversity and abundance of leaf architectural types of fossil floras. Differences in TCT proportions and quantitative leaf traits may be consistent with local variations in the proportion of broad-leaved deciduous and evergreen elements in the ecotonal vegetation of the early Oligocene. A correlation between leaf size, LMA, and fossil-species indicates that trait variations are partly dependent on the taxonomic composition. Leaf morphology or TCTs itself cannot explain the difference in insect herbivory on leaves. It is a more complex relationship where leaf morphology, LMA, phenology, and taxonomic affiliation are crucial.

Introduction

Fossil leaves are known as archives for paleoenvironmental parameters. As primary photosynthetic organs, leaves are directly exposed to their environment. Specific morpho-anatomical leaf characteristics (from now on, traits) reflect plant adaptations or responses to habitat conditions (Violle et al., 2007; Díaz et al., 2016; Moraweck et al., 2019; Li et al., 2022). Measuring traits on fossil leaves allows inferences about the paleoecological context when assemblages from several localities/ages are compared (e.g., comparatively warmer or drier conditions; Roth-Nebelsick et al., 2017). Two essential traits are the leaf surface (leaf area) and the leaf mass per area (LMA), which may reflect water availability and temperature differences, and plant conservative/acquisitive strategies (Wright et al., 2017; Peppe et al., 2018). LMA is determined indirectly for fossil leaves by an equation that uses the leaf area and the petiole width at its leaf insertion point (Royer et al., 2007). The positive correlation of LMA with leaf life span indicates the leaf economics of fossil-species (Reich, Walters & Ellsworth, 1997; Westoby et al., 2002; Wright & Westoby, 2002; Wright et al., 2004). Therefore, it is applied in paleoecology to distinguish between deciduous (fast return strategy with low resource investment into leaf structure) and evergreen (slow return strategy with high resource investment into leaf structure) leaves. Furthermore, a multi-trait approach was proposed by Roth-Nebelsick et al. (2017) that defines 16 trait combination types (TCTs) of dicot leaves based on their general morphological characteristics. Recording these TCTs for specimens of a fossil leaf assemblage allows for documenting the morphological spectrum, which might provide additional information about environmental constraints in an integrative way (Roth-Nebelsick et al., 2017). Indeed, although informed categorically in the TCT approach, variations in these traits were associated with environmental characteristics (e.g., a higher proportion of leaves with toothed margins occurs in colder environments; Peppe et al., 2018).

Moreover, different combinations of structural and chemical traits cause different types of leaves to have different susceptibilities to insect herbivory (e.g., Coley & Barone, 1996; Knepp et al., 2005; Pringle et al., 2011; Nakamura, Inari & Hiura, 2014; Silva, Espírito-Santo & Morais, 2015; Nascimento et al., 2019). Leaf structural traits, such as the density of leaf veins and pubescence, or mechanical properties, such as the toughness and strength of leaf structural elements, can negatively impact insect herbivores by increasing the time and energy required for ingestion and digestion. In the case of small insect herbivores, these traits affect by preventing access to the leaf surface (e.g., Choong et al., 1992; Lucas et al., 2000; Read & Stokes, 2006; Hanley et al., 2007; War et al., 2012). Overall, the effect of these physical defenses differs significantly among insect herbivores and can be overcome by specific adaptations (Schoonhoven, van Loon & Dicke, 2005). The same applies to modern plants’ broad spectrum of chemical plant defenses. These are, however, practically unknown for fossil plants.

Over the last decades, paleoecological studies have increasingly considered exploring the interaction between plants and other organisms in ancient ecosystems. Fossil leaves were examined for the presence and frequency of damage types caused by herbivorous arthropods (most probably insects) or fungi (e.g., Wilf et al., 2001, 2006; Currano, Labandeira & Wilf, 2010; Wappler et al., 2012; Knor et al., 2013; Donovan et al., 2016; Müller, Wappler & Kunzmann, 2018; Azevedo Schmidt et al., 2019; Adroit et al., 2021; Currano et al., 2021; Labandeira, 2021; Maccracken et al., 2021; Schachat, Payne & Boyce, 2021; Maccracken et al., 2022; Santos et al., 2022). Various metrics, such as damage frequency, herbivory index (i.e., the ratio between damaged and undamaged leaf surface; Schachat, Labandeira & Maccracken, 2018; Schachat, Maccracken & Labandeira, 2020), or damage type richness, were developed to describe, quantify, and compare herbivory patterns among fossil assemblages or fossil-species. In addition, they were used to infer the presence and diversity of herbivorous insects indirectly, trace temporal and spatial herbivory variations, or reconstruct parts of the trophic network and ecosystem structure (e.g., Currano et al., 2021; Schachat, Payne & Boyce, 2021; Swain et al., 2021).

Combining the TCT approach, quantitative leaf traits, and insect herbivory metrics has yet to be explored. However, an integrative approach when analyzing fossil floras and combining the source of information is always better for reliable reconstructions. It may contribute to our knowledge of plant-insect interactions in ancient ecosystems. Here, dicot leaf assemblages of two early Oligocene fossil floras were studied: Seifhennersdorf (Sf, Saxony, Germany) and Suletice-Berand (SuBe, Ústí nad Labem Region, Czech Republic). The assemblages were selected because (1) they are similar in age, paleoclimate, depositional setting, and paleogeographic position (Table 1), (2) they consist of numerous taxonomically determined leaves of diverse fossil floras (Kvaček & Walther, 1995; Walther & Kvaček, 2007), (3) they were assigned to the same vegetation type by Integrated Plant Record vegetation analysis, namely the ecotone between broad-leaved evergreen and mixed mesophytic forest (Teodoridis & Kvaček, 2015), (4) their preservation allows for collecting qualitative and quantitative leaf trait data, and (5) they show traces of insect herbivory on the leaves. The first extensive but not published analysis of insect damage types from the Seifhennersdorf flora was conducted by Koch (2011). For the present contribution, the determinations and analysis were revised. So far, neither insect body fossils nor insect damage types on leaves have been investigated from Suletice-Berand.

Table 1 Key data of the fossil floras: topographical, geological, environmental, and paleobotanical information.

	Seifhennersdorf	Suletice-Berand	
Topography			
Location	Saxony, Germany	Ústí nad Labem Region, Czech Republic	
UTM-Coordinates	33U 473877 5641714	33U 440433 5610423	
Geology			
Sediment type	Lacustrine diatomitic pelite	Lacustrine diatomitic pelite	
Stratigraphy	Middle lower Oligocene	Upper lower Oligocene	
Age	30.7 +/− 0.7 Ma
K-Ar of basaltoid rock above
diatomitic pelite series
(Bellon et al., 1998)	26.0–29.0 Ma
(Teodoridis & Kvaček, 2015)	
Floristic analysis			
Systematic description	Walther & Kvaček (2007)	Kvaček & Walther (1995)	
Floristic complex	Seifhennersdorf-Kundratice
(Kvaček & Walther, 2001)	Nerchau-Flörsheim
(Kvaček & Walther, 2003)	
Integrated Plant Record	Ecotone between broad-leaved evergreen and mixed mesophytic forest
(Teodoridis & Kvaček, 2015)	Ecotone between broad-leaved evergreen and mixed mesophytic forest
(Teodoridis & Kvaček, 2015)	
Paleoclimate			
Analysis	Moraweck et al. (2019)	Teodoridis & Kvaček (2015)	
Köppen-Geiger type	Cfa	Cfa	
			
Coexistance Approach			
MAT (°C)	16.5–18.3	15.6–18.3	
WMMT (°C)	25.6–25.9	24.7–27.5	
CMMT (°C)	9.0–10.9	5.0–10.9	
MAP (mm)	1,231–1,333	1,096–1,213	
MPwet (mm)	180–191	N/A	
MPdry (mm)	43–51	N/A	
			
CLAMP			
MAT (°C)	10.0 (1.3)	12.4 (1.2)	
WMMT (°C)	22.9 (1.7)	24.8 (1.4)	
CMMT (°C)	1.6 (2.6)	1.6 (1.9)	
GSP (mm)	849.4 (497)	N/A	
3_wet	559.2 (239)	71.6 (13.8)	
3_dry	140.4 (104)	18.6 (3.2)	
Note:

MAT, Mean annual temperature; WMMT, Warm month mean temperature; CMMT, Cold month mean temperature; MAP, Mean annual precipitation; MPwet, Mean precipitation of the wettest months; MPdry, Mean precipitation of the driest months; CLAMP, Climate Leaf Analysis Multivariate Program; GSP, Growing season precipitation; 3_wet, Mean precipitation of the three wettest months; 3_dry, Mean precipitation of the three driest months; Cfa, Humid subtropical climates (C = warm temperate, f = fully humid, a = hot summer); N/A, Not available.

This study presents the Integrated Leaf Trait Analysis (ILTA), a workflow for the combined application of methodologies in TCT, quantitative leaf trait, and insect herbivory analyses on dicot leaf assemblages. The objectives were (1) to record the leaf morphological variability among sites, (2) to describe the characteristics of insect herbivory patterns, (3) to explore relations between TCTs, quantitative leaf traits, and fossil-species ecological characteristics inferred from taxonomic knowledge (e.g., growth form, phenology), and (4) to explore possible relations of selected leaf characteristics and insect herbivory.

Materials and Methods

Geological background and fossil site features

Volcanic activity during the Paleogene to Neogene in Northern Bohemia (Czech Republic) and the Upper Lusatia in southeast Saxony (Germany) was caused by tectonic movements in the ENE–WSW trending Ohře (Eger) Graben, the eastern part of the European Cenozoic rift system (Ziegler, 1992, 1994). The Ohře Graben is characterized by volcanic complexes, wedged brown coal basins, and by the formation of marginal depressions (Akhmetiev, Walther & Kvaček, 2009). During the early Oligocene, such a local depression with a paleo-lake was formed between the localities of Seifhennersdorf (Saxony, Germany) and Varnsdorf (Ústí nad Labem Region, Czech Republic). The lake infilling is recorded by a up to 60 m thick volcano-sedimentary sequence that contains interbeds of fossiliferous diatomite seams (Ahrens, 1959; Walther & Kvaček, 2007). The sequence ends with a potassium-argon-dated basaltic lava flow of 30.7 ± 0.7 million years (Bellon et al., 1998). Overall, five seams of lacustrine diatomite developed in temperate humid but seasonal paleoclimate (Table 1, Schiller, 2007). Seams four and five, in the lowest part of the sequence, reached the highest thicknesses (4 and 10 m) and contained rich fossil fauna and the Seifhennersdorf flora (Walther & Kvaček, 2007). Volcanism and tectonic movements controlled the lake’s evolution and shaped the morphology of the lake’s surroundings. That caused the development of various habitats inhabited by diverse plant communities at the shore of the Seifhennersdorf lake and within its proximity (Walther & Kvaček, 2007). Mixed mesophytic forest with, e.g., Acer, Betula, Carpinus, Carya, Daphnogene, Dombeyopsis, Laurophyllum, Platanus, Rosa, Sloanea, Ulmus, and Zelkova developed on hilly non-waterlogged sites whose soils derived from deeply weathered Lausitz Granodiorite. Flat and periodically flooded lake shores favored riparian forest development with, e.g., Alnus, Carpinus, Carya, Eotrigonobalanus, and Populus on fertile soils of alkaline volcanic rocks. Local rather permanently flooded depressions favored the colonization by a Taxodium-dominated swamp forest (Walther & Kvaček, 2007).

The fossil site of Berand near the village of Suletice, called Suletice-Berand, is situated in the east of Ústí nad Labem in Northern Bohemia (Czech Republic) within the České středohoří Mountain’s volcano-sedimentary complex (ČsMts. complex). The ČsMts. complex, as a geological unit of the Ohře Graben, is characterized by a lithostratigraphy showing a polygenetic assemblage of alkaline superficial and intrusive volcanism, pyroclastics, and accompanying sedimentary intercalations (Cajz, 2000; Akhmetiev, Walther & Kvaček, 2009). The fossil flora of Suletice-Berand is preserved in freshwater diatomite embedded in a sequence of basaltoid pyroclastics of the Ústí Formation (Cajz, 2000). Contrary to Seifhennersdorf, knowledge about local geology, stratigraphic position, and the structure of the paleo-lake, most probably the infilling of a volcano-tectonic depression, are limited and only partly clarified by studies of core drillings and in test pits (Kvaček & Walther, 1995; Walther, 2004). Paleobotanical investigations indicate the lack of swampy and riparian habitats like in Seifhennersdorf. Non-waterlogged areas were inhabited by a mixed mesophytic forest characterized by, e.g., Acer, Daphnogene, Engelhardia, Laurophyllum, Leguminosites, Platanus, Sloanea, and Zelkova (Kvaček & Walther, 1995; Akhmetiev, Walther & Kvaček, 2009). Based on the Integrated Plant Record vegetation analysis, the floras of Seifhennersdorf and Suletice-Berand are assigned to a vegetation type defined as an ecotone between broad-leaved evergreen and mixed mesophytic forest (Teodoridis & Kvaček, 2015). Additional geological background and fossil site information are provided in Article S1.

Fossil leaf collections

The specimens investigated are being kept in the paleobotanical collection of the Senckenberg Natural History Collections Dresden (Germany, Museum of Mineralogy and Geology) under the identification MMG PB Sf (Seifhennersdorf) and MMG PB SuBe (Suletice-Berand). A list of specimens, including their accessory numbers, is available in Table S1. The collection history and collecting procedures of the studied specimens are different. Article S1 contains supplemental information on this. For the present study, 4,935 leaves from Seifhennersdorf and 1,349 from Suletice-Berand were analyzed.

The leaves preserved in lacustrine diatomite were most likely complete during their embedding. Fragmentation of fossil leaves was caused by the sampling process (sediment split off) and the limit of hand-sized specimens. Most sampled leaves are preserved in at least 50%. However, the preservation of leaf laminae is distinctly different between specimens from each fossil site. The Seifhennersdorf specimens were affected by the chipping of coalified/compressed material caused by the desiccation of the thin coaly leaf laminae under storage conditions in the collection room over decades. Possible limitations arising from this are discussed in the methodology sections below. The leaves from Seifhennersdorf are preserved as compressions (i.e., coalified leaves with cuticles preserved). The leaves of Suletice-Berand are altered, resulting in the decay of the coalified matter and cuticles. Therefore, only Seifhennersdorf specimens are at least partly determined by additional micromorphological characters of leaf cuticles (Walther & Kvaček, 2007).

Workflow

The Integrated Leaf Trait Analysis includes the following steps and methods (Fig. 1).

Figure 1 Schematic overview of the datasets used and their selection process.

Datasets in bold represent the datasets used for analyses in this study. Datasets in light represent necessary intermediate steps in sampling or represent a dataset for an analysis whose results are given in the supplemental information (Dataset 4). *Dataset 3 cleaned for specimens having leaf area and LMA outliers, incomplete data, and are of rare fossil-species/TCTs (less than five leaves). **Completeness of observations of variables is mandatory for the applied multivariate analysis. In the present study, the presence of LMA values was limiting.

Performing the sampling

1) Dicot angiosperm leaves that are taxonomically determined or have at least sufficiently well-preserved morphological characteristics have been selected for studying TCTs and insect damage types (Dataset 1). At least 50% of the leaf laminae should be preserved.

2) Leaves at least 70% preserved were documented (Dataset 2), of which 400 leaves per site were then randomly selected for the graphical reconstruction of the leaf outline and the determination of quantitative leaf traits (Dataset 3).

3) For each sampling step, the taxonomic composition was documented (Fig. 2).

4) Multivariate analyses were performed on subsets of Dataset 3 having no missing values (Datasets 4 and 5). A description of the datasets is available in Table 2.

Figure 2 Differences in taxonomic composition between leaf assemblages and different datasets.

(A) Frequency of plant families occurring in the Seifhennersdorf assemblage and their representation in the datasets. (B) Frequency of plant families occurring in the Suletice-Berand assemblage and their representation in the datasets. The changes in frequency document the effect of the sampling process described in Material & Methods.

Table 2 The different datasets used with description, mean leaf size, and LMA.

Differences between Seifhennersdorf and Suletice-Berand in leaf size and LMA are indicated by <0.05 p-values (Mann-Whitney test).

Data	n	Mean leaf size (mm2)	Mean LMA (g/m2)	
Dataset 1
Whole sample of leaves.	6,284
(Sf: 4935,
SuBe: 1349)	–	–	
Dataset 2
Leaves with ~70% of lamina preserved.	1,976
(Sf: 1556,
SuBe: 420)	–	–	
Dataset 3
All leaves having quantitative trait measurements.	800
(Sf: 400,
SuBe: 400)	Sf = 1262.40 (SD = 963.10)
SuBe: 766.73 (SD = 786.24)
W = 111396, p-value < 2.2e−16	Sf = 84.92 (SD = 23.34; NAs = 225)
SuBe = 104.84 (SD = 29.97; NAs = 257)
W = 7397, p-value = 3.582e−10	
Dataset 4*
All leaves from Dataset 3 minus specimens with an outlier (size/LMA), of rare fossil-species and TCTs (<5), and with incomplete cases.	228
(Sf: 124,
SuBe: 104)	Sf = 1104.40 (SD = 730.06)
SuBe = 452.90 (SD = 476.26)
W = 10626, p-value < 2.2e−16	Sf = 82.92 (SD = 19.64)
SuBe = 102.70 (SD = 24.93)
W = 3425, p-value = 1.11e−09	
Dataset 5*
Contains specimens of Dataset 4 with measured LMA and additional specimens with simulated LMA.	506
(Sf: 244,
SuBe: 262)	Sf = 1064.28 (SD = 704.95)
SuBe = 526.27 (SD = 437.16)
W = 50377, p-value < 2.2e−16	Sf = 81.62 (SD = 20.91)
SuBe = 103.78 (SD = 24.02)
W = 15979, p-value < 2.2e−16	
Notes:

* Datasets 4 and 5, the two subsets of Dataset 3, are used for multivariate analyses (see Material & Methods). They contain the following data: leaf size, length/width ratio, length, width, LMA (calculated), TCT (specimen-based, taxonomy-based, and combined approach), area index, phenology, and growth form.

n, Number of leaves; SD, Standard deviation; LMA, Leaf mass per area; Sf, Seifhennersdorf assemblage; SuBe, Suletice-Berand assemblage; W, Statistic of the Mann-Whitney-Wilcoxon test; NAs, Non-available values.

The qualitative specimen data are available in Table S2 and the quantitative in Table S3.

Determining the trait combination types

The TCT approach by Roth-Nebelsick et al. (2017) categorizes dicot leaf architectural types in an assemblage. The classification of fossil leaves is made by using a hierarchic scheme based on four morphological characteristics: (1) lobation (lobed vs. unlobed), (2) leaf margin type (entire margins vs. toothed margins), (3) primary venation type (pinnate vs. palmate primary venation), and (4) secondary venation type (looped vs. non-looped secondary venation). In total, 16 TCTs are possible, each named with a capital letter from A to P (Fig. 3). The morphological characteristics were determined following the Manual of Leaf Architecture (Ellis et al., 2009). The assignment of a TCT to a specimen was made from direct observations of the characters as much as possible (specimen-based TCT approach; Roth-Nebelsick et al., 2017). In cases of poor preservation or visibility of secondary venation characteristics for specimens of well-determinable fossil-species (e.g., Engelhardia orsbergensis, Platanus neptuni, or Rosa lignitum), TCTs were determined based on the fossil-species diagnoses (combined specimen- and taxonomy-based TCT approach; Kunzmann et al., 2019). The specimen-based, taxonomy-based, and combined TCTs are stated in Table S2. Specimens in which TCTs could not be determined directly or by taxonomy (e.g., in cases where the fossil-species diagnosis implies multiple possible TCTs) were excluded from the analysis. The chipping of the coalified material at the Seifhennersdorf specimens did not affect the TCT determinations (e.g., characters are visible in better-preserved areas of the lamina or from the impression in oblique light). Poor visibility of secondary venation types was usually related to very dark coalified leaf laminae.

Figure 3 Frequency of trait combination types (TCT) compared between Seifhennersdorf and Suletice-Berand.

The figure is based on Dataset 1 and shows the results of a combined specimen- and taxonomy-based TCT approach. For details, see Material & Methods.

Reconstructing the leaf outlines

Because the lamina of fossil leaves is often incompletely preserved, digital reconstruction techniques to replenish missing parts of the leaf lamina outline are necessary before quantitative traits can be measured. A reliable reconstruction is based on the taxonomic description of a fossil-species and its morphological variability. According to Moraweck et al. (2019), the replenishment of leaf lamina should be at most 30% (meaning at least 70% of the original leaf lamina should be preserved). The ratio between the preserved and replenished leaf lamina is given by the area index (IA) or preservation index (Traiser et al., 2018; Moraweck et al., 2019). Image acquisition, processing, and graphical reconstruction were made according to Traiser et al. (2018). The 400 selected leaves per site (Dataset 3) were photographed with a Canon Power Shot G1 X Mark III. After image processing, the minimum (preserved) and maximum (replenished) leaf lamina outlines were generated as separate layers and saved in a shapefile format using the open-source geographic information system QGIS (version 3.10.14-A Coruña and 3.12.1-Bucuresti; QGIS Development Team, 2020). The method allows storing leaf lamina reconstructions, checking reliability, and ensuring data reproducibility.

Measuring the leaf quantitative traits

The maximum leaf lamina outlines were used for measurements of leaf length (in mm), leaf width (in mm), and leaf area (in mm2). The leaf length/width ratio was calculated. LMA (in g/m2) was calculated by using the Eq. (1) for dicot leaves by Royer et al. (2007). Because this formula requires measurements of the petiole width (in mm) at the insertion point of the lamina and that petioles are often not preserved, LMA could only be retrieved for 175 fossil leaves from Seifhennersdorf and 143 fossil leaves from Suletice-Berand (Table S3). Means for leaf area and LMA were calculated for a fossil-species with at least five leaves.

(1) log(LMA)=3.070+0.382×log(petiolewidth2/fossilleafarea)

Inferring the fossil-species phenology

Since the taxonomic affiliation of most fossil-species in the studied assemblages is known (Kvaček & Walther, 1995; Walther & Kvaček, 2007), comparisons with modern relatives were used to infer the phenology of the fossil-species and thus to distinguish between evergreen and deciduous. Additionally, for fossil-species with at least five measurable leaves (Tables S4 and S5), the mean LMA was used as an approximation of phenology as introduced by Royer et al. (2007): LMA < 87 g/m2 = deciduous, LMA between 87 and 129 g/m2 = intermediate, and LMA > 129 g/m2 = evergreen. Phenology information derived from modern relatives was compiled and used in multivariate analysis to assess the congruence of these estimates and the LMA data.

Determining the insect damage types and herbivory metrics

First, the leaves in Dataset 1 were scored for the presence or absence of insect herbivore traces, further called insect damage types (DTs). Each DT was classified using Labandeira et al. (2007) DT catalog. A short description and an identification number define a DT. Further, DTs are assigned to terrestrial arthropod functional feeding groups (FFGs), such as external foliage-feeding, leaf-mining, or galling. Three criteria are used to distinguish DTs from detritivory or mechanical destruction: (1) the presence of distinctive, thickened tissues (callus), (2) structural features of herbivore-induced alterations such as resistant veinal stringers, necrotic tissue flaps or areas of secondary fungal infection and (3) distinctive stereotypy of a feeding pattern on a particular fossil-species (Labandeira, 2013). As mentioned above, the Seifhennersdorf specimens were affected by chipping coalified/compressed material. The extent of chipping varies from specimen to specimen and may also have affected the herbivory results. External foliage-feeding DTs were securely determinable since they were also recognizable as imprints. A loss of galls and leaf mines due to the chipping of material cannot be ruled out, which could reduce these FFGs’ occurrences and diversities. Therefore, herbivory metrics are probably partly affected by the state of leaf preservation. However, the scale is difficult to estimate.

Second, the sample-based metrics damage frequency, DT occurrence, and DT richness were used to describe, quantify, and compare insect herbivory patterns. The damage frequency (DAM%) represents the proportion between damaged and undamaged leaves in a sample (query DTs present or absent). The DT occurrence (DTO) states a sample’s number of DTs (absolute frequency). Since a leaf can have multiple occurrences, we did not standardize this metric by the number of leaves. The DT richness (DTRnumber of leaves) reports the number of different DTs in a sample. Sample-based rarefaction procedures (Gotelli & Colwell, 2001) were applied using the free statistical software R (R Core Team, 2020) to standardize DT richness and ensure comparability between the fossil sites. In detail, we used the extension of analytical rarefaction by Gunkel & Wappler (2015). The total, external foliage-feeding (hole-feeding, leaf margin-feeding, leaf surface-feeding, leaf-skeletonization), leaf-mining, and galling DT richness were computed. The standard deviation (SD) for the resamples was calculated. The DT matrices for calculating the figured rarefaction curves are provided as supplemental information (Data S1–S6). The three metrics were applied to the two entire leaf assemblages and fossil leaves with assigned TCTs, fossil-species phenology (deciduous, evergreen, or semi-deciduous), and taxonomic affiliation (Dataset 1). The cut-off for rarefaction was 20 leaves.

Third, for the specimens showing DTs in Dataset 3, the damaged area (in mm2) was measured and related to the fossil leaf area (in mm²) to determine the herbivory index (HI). Damaged area and HI were used as specimen-based herbivory metrics for generalized linear model analyses in addition to DTs’ presence/absence data.

Performing the statistical analyses

Several statistical uni- and multivariate analyses were made using R 4.0.3 (R Core Team, 2020) to (1) test for the significance of differences in single leaf traits between Seifhennersdorf and Suletice-Berand, to (2) describe leaf characteristics from multiple quantitative traits and highlight possible trait co-variation (principal component analysis, PCA) and (3) test for the relationship between specific leaf characteristics (e.g., quantitative leaf traits and insect herbivore traces) with assemblage and vegetation properties (several generalized linear models, GLM). The R script is available in the supplemental information (Data S7).

For univariate analysis, Shapiro-Wilk tests were used to evaluate the data normality (R Core Team, 2020). This assumption being often not met in our data, differences for specific leaf traits were tested with the non-parametric Mann-Whitney test (R Core Team, 2020). Kruskal-Wallis post-hoc tests were used to compare multiple group values (e.g., leaf size or LMA variation among plant families and TCTs; function “kruskal”, Package agricolae; Mendiburu, 2020). Quantitative leaf traits were analyzed with a PCA (i.e., five traits: LMA, leaf size, leaf width, leaf length, leaf length/width ratio) using the packages FactoMineR (Lê, Josse & Husson, 2008) and factoextra (Kassambara & Mundt, 2020). This method is commonly used to characterize the functional diversity of present-day vegetation and is frequently applied in paleontological research (e.g., Díaz et al., 2016; Segrestin et al., 2021). Morphospaces were computed based on the two first principal components to illustrate leaf morphological disparity among the specimens of both assemblages, taxonomic units, and TCTs (Fig. 4). Five GLMs were made to highlight the possible relationship of leaf size (M1), LMA (M2), and herbivory interaction (presence/absence of interaction (2), the area damaged (4), and HI (5)) with ecological parameters (i.e., locality, taxonomy, phenology, growth form, and TCTs), preservation (IA), and for herbivory interactions, leaf traits. Plant-insect interactions were modeled in two steps, with a first GLM exploring the probability of showing DTs (presence or absence of interactions, binomial model) and then two GLMs investigating the relationship of the leaf surface damaged with leaf and assemblage properties, using two different metrics: the area damaged or HI. All GLMs were parametrized with a Gamma distribution and identity link, except for M3 being a binomial model (link = logit). The best-fitting models, whose formulas are given in the results, were selected based on the Akaike Information Criterion (AIC, function “step,” package Stats; R Core Team, 2020). For these, the significance of explanatory variables was tested with a type-II ANOVA (function “Anova,” package car; Fox & Weisberg, 2019), whose validity was checked by investigating normality and homogeneity of the residuals with the function “check_model” (package performance; Lüdecke et al., 2021). R-squared were obtained to assess the different model’s explanatory power; they correspond to Nagelkerke’s R2 or Tjur’s R2 (for the binomial presence/absence model, M3; function “R2”, package performance; Lüdecke et al., 2021).

Figure 4 The morphospace of leaves is built from the two first axes of the Principal Component Analysis.

(A) Points location reflects leaf quantitative trait variability. (B–D) Highlight the possible relationship with environmental/ecological variables. Only phenology significantly affects leaf morphology (i.e., as assessed with the GLMs on leaf size and LMA, M1 and M2). The PCA was made on data after LMA data imputation (Dataset 5). For details, see Material & Methods.

The presence of missing data in a dataset significantly reduces the number of specimens used in multivariate analyses (i.e., observations for all selected explanatory variables are mandatory). Within our data, the low availability of LMA estimates was strongly limiting (i.e., it was only possible to calculate it for the 228 leaves with their petioles preserved over the 800 graphical reconstructed leaves). Therefore, multivariate analyses were computed on two datasets (Fig. 1 and Table 2): a first one, containing all specimens having calculated LMA values (n = 228 leaves, Dataset 4), and a second dataset, an enlarged version of the first one, for which some of the missing LMA values could be simulated (n = 506 leaves, Dataset 5). LMA completion was made by using multivariate imputation by chained equations (function “mice” of the package MICE; Van Buuren & Groothuis-Oudshoorn, 2011) with the method “norm” (Bayesian linear regression). This method has the advantage of not biasing the variance (contrary to the common use of species mean values; Van Buuren & Groothuis-Oudshoorn, 2011). Note that it was not possible to simulate LMA values from regression models because the number of specimens per fossil-species was too low, and no sufficiently strong correlation (R2 > 0.4) was found between LMA and other continuous variables per taxa (fossil-species as genus being considered, not shown). Missing LMA values were simulated for leaves belonging to a fossil-species with at least five calculated LMA values in the respective assemblage. The leaves’ locality and size were indicated as explanatory variables in the simulation process. Each predicted LMA value averages 10 values (10 imputation cycles). Different analyses were made to assess the imputation quality (Fig. S1).

For both datasets, leaf size or LMA outliers (> or <1.5 times the interquartile distance) and observations from fossil-species with less than five leaves were removed to avoid bias in multivariate analyses (false positives). Similarly, rare TCTs (less than five observations) were clustered or deleted. A significance level, α, of 0.05 was used for all analyses. Because predicted LMA values are consistent with the observed values (Article S2), presented are the results obtained from the dataset with partly simulated LMA values (Dataset 5). That allowed the inclusion of many more specimens in the multivariate analyses.

Results

Leaf morphological-morphometric variability

Leaf morphological characteristics and TCTs

Seifhennersdorf leaves (n = 4,935) are mainly unlobed (88.89%) and toothed (82.53%), have pinnate primary venation (87.50%) and non-looped secondary venation (74.14%). Suletice-Berand leaves (n = 1,349) are mainly unlobed (94.51%) and toothed (58.49%), have pinnate primary venation (92.06%) and more frequently looped secondary venation (64.05%). As shown in Fig. 3, the assemblages are dominated by five TCTs (>20 leaves): A, B, E, F, and P. Differences between the assemblages exist in the TCT proportion. The Seifhennersdorf assemblage is characterized by a high frequency of TCT F leaves (64.78%), coming from diverse deciduous trees and shrubs like Ailanthus prescheri, Alnus phocaeensis, Betula alboides, Carpinus grandis, Carpinus roscheri, Carya fragiliformis, Cyclocarya sp., Ostrya atlantidis, Populus zaddachii, Salix varians, Ulmus fischeri and Zelkova zelkovifolia (Table S4). Leaves of TCTs E (8.17%), P (6.36%), A (3.55%), and B (2.84%) are less frequent. TCT E is mainly related to Platanus neptuni and Rosa lignitum, TCT P to fossil Acer species, TCT A to evergreen trees and shrubs like Laurophyllum acutimontanum and Magnolia seifhennersdorfensis, and TCT B to diverse legume morphotypes and Cornus sp. Twenty fossil-species show leaves of TCT F, ten of TCT E, nine of TCT A, six of TCT P, and a minimum of two fossil-species shows TCT B (e.g., it is uncertain how many fossil-species are represented by the legume morphotypes).

TCTs E (40.62%) and A (21.50%) represent most leaves with looped secondary venation in Suletice-Berand. Conversely, leaves with non-looped secondary venation are less frequent in that assemblage (TCT F: 12.08%, TCT P: 3.56%), except leaves of TCT B (11.19%) that are frequently represented by legume morphotypes (Fig. 2). Leaves of TCT E often come from Engelhardia orsbergensis, an evergreen tree, and less frequently from P. neptuni, a deciduous tree. Leaves of TCT A are from smaller evergreen trees to shrubs like Laurophyllum acutimontanum, L. pseudoprinceps, and Oleinites maii (Table S5). Like TCT E, in Suletice-Berand, leaves of TCT F can either belong to evergreen (e.g., Sloanea olmediifolia) or deciduous fossil-species (e.g., Carpinus grandis, Carya serrifolia, or Zelkova zelkovifolia)—with evergreens predominating proportionally (Table S4). Lobed and palmately-veined leaves (TCT P) are from fossil Acer species, too. Thirteen fossil-species are included in TCT F, 11 in TCT A, eight in TCT E, four in TCT P, and a minimum of three in TCT B. Both assemblages contain TCTs that are represented by less frequently preserved fossil-species (less than 20 leaves) like Cercidiphyllum crenatum (TCT G), Tilia gigantea (TCT H), and cf. Crataegus sp. (TCT M) in Seifhennersdorf or Pungiphyllum cruciatum (TCT J) and Dombeyopsis sp. (TCT K) in Suletice-Berand (Fig. 3).

Leaf quantitative traits

Based on quantitative trait analyses (Dataset 3), leaves tend to be larger and with a lower LMA in Seifhennersdorf than in Suletice-Berand. These differences in leaf size and LMA are also significant in Dataset 5, used for multivariate analyses (Table 2). The calculated mean LMA of seven fossil-species from Seifhennersdorf is below 87 g/m2 (Fig. 5 and Table S4)—the threshold for deciduousness stated by Royer et al. (2007). The calculated values are consistent with inferred phenology from modern relatives. In Suletice-Berand, the calculated mean LMA of six fossil-species falls into the transition between deciduous and evergreen (i.e., between 87 and 129 g/m2, Royer et al., 2007; Fig. 5 and Table S5). These LMA values are lower than expected from the phenology information inferred from modern relatives, which suggest Daphnogene cinnamomifolia, Engelhardia orsbergensis, Laurophyllum cf. acutimontanum, and Symplocos deichmuelleri most likely to be evergreen trees or shrubs. The opposite is observed for Platanus neptuni and Zelkova zelkovifolia, which were most probably deciduous by their modern relatives but showed mean LMA greater than 87 g/m2 (Fig. 5 and Table S5).

Figure 5 Leaf size and leaf mass per area (LMA) variabilities.

(A) Leaf size per family. (B) LMA per family. Red dots represent the Seifhennersdorf specimen. Blue dots represent the Suletice-Berand specimen. Grey dashed lines show the LMA boundaries that Royer et al. (2007) defined: LMA < 87 g/m2 = deciduous, LMA between 87 and 129 g/m2 = intermediate, and LMA > 129 g/m2 = evergreen. Letters highlight groups of families sharing the same size or LMA (i.e., families having a letter in common are not significantly different; Multiple Kruskal tests, R package agricolae, p-value <0.05). For details, see Material & Methods.

Multivariate analyses of leaf quantitative traits

Results of multivariate analyses (PCA and GLM) are derived from Dataset 5, which contains calculated and simulated LMA values (Fig. 1). Figures showing the quality of the LMA imputation are available in the supplemental information (Fig. S1). The same analyses were also conducted on the smaller Dataset 4 (without imputing LMA missing values) and are available in Article S2.

The morphospace obtained from the PCA on quantitative leaf traits is illustrated in Fig. 4. The two first axes, namely PC1 and PC2, explain 83.9% of the variance (54.2% and 29.7%, respectively). PC1 indicates the leaves’ size, width, length, and LMA (each parameter explaining 33.1%, 28.7%, 18.1%, and 16.4% of specimen distribution along the axis, respectively). Thereby, small-sized leaves with higher LMA are associated with negative values. In contrast, large-sized leaves with lower LMA but higher widths and lengths are associated with positive values. PC2 mainly reflects circularity (l/w ratio explains 57.6% of point distribution along the axis), with leaves of high (resp. low) l/w ratio associated with positive (resp. negative) values.

To better understand leaf area and LMA variabilities among leaves, GLM analyses were run. Among the variables tested, those that best explain leaf size and LMA variations are contained in the following models:

M1:LMA~Species+leafsize+phenology+IA(+locality),

M2:Leafsize~Species+IA+phenology(+locality).

The parameters indicated after “~” are the significant explanatory variables as given by the function “step” (see Material & Methods). The variables which best explain LMA variations (model M1) are the taxonomic affiliation (fossil-species), leaf size, and phenology (Table S6). LMA is negatively correlated to leaf size but positively associated with evergreen and, to a lesser extent, semi-deciduous species (Figs. 4A and 4C). A positive relationship with leaf preservation (IA) is also observed, leaves of higher LMA tend to be more complete. This could be due to a higher preservation potential of these leaves than leaves with low LMA (i.e., lower fragmentation before fossilization and during the fossil collection). Similarly, leaf size variations are also partly explained by taxonomic affiliation (fossil-species), leaf preservation (IA), and phenology (model M2). Negative relationships are observed between leaf size, leaf preservation, and phenology (i.e., the smallest leaves are the best preserved and are related to evergreen fossil-species).

The significant effect of fossil-species on leaf area and LMA shows that the taxonomic composition of assemblages is important to consider during sampling (Fig. 2) and to study these traits at a paleo-community level. Thus, part of the differences in leaf size and LMA between Seifhennersdorf and Suletice-Berand likely result from the different fossil-species compositions and proportions in the two assemblages (Fig. 4B). Some families with the largest specimens are more abundant in Seifhennersdorf, especially Sapindaceae and Betulaceae. Conversely, families with smaller specimens are more represented in Suletice-Berand (e.g., Fabaceae, Juglandaceae, Lauraceae, and Platanaceae; Fig. 5A). Similarly, families with specimens tending to higher LMA (e.g., Fabaceae, Lauraceae, Platanaceae, Symblocaceae, and Ulmaceae) are more abundant in Suletice-Berand than in Seifhennersdorf (Fig. 5B). While significant differences were found when testing the correlation between size or LMA and locality alone (Table S6), the locality was not found to be a significant parameter in models M1 and M2 (p-values of 0.14 and 0.09, respectively). Interestingly, this shows that the locality factor does not explain most of the variation in size (or LMA) between specimens and highlights the interest in multivariate approaches to understand trait variations. The results are fairly consistent with those obtained on the smaller Dataset 4 without LMA imputation (Fig. 1), although no relationship was found between leaf preservation (IA) and LMA (Article S2).

Insect herbivory pattern

General comparison

Herbivory metrics compared between the assemblages (Dataset 1) indicate that insect herbivory was more pronounced on leaves from Suletice-Berand (Fig. 6). Representative examples of DTs are shown in Figs. 7 and 8. In detail, 4.78% of 4,935 leaves from Seifhennersdorf and 12.08% of 1,349 leaves from Suletice-Berand were affected by insect damage. The DT occurrences between the assemblages are compared in Fig. 6B1. Regarding the number of sampled leaves, DT occurrence was denser in the Suletice-Berand assemblage. Moreover, the richness of DTs, standardized by sample-based rarefaction, is more significant in Suletice-Berand than in Seifhennersdorf (Fig. 6C1). On 400 sampled Seifhennersdorf leaves 2.59 FFGs (SD = 0.67) and 7.92 different DTs (SD = 1.68) are present, whereas 4.60 FFGs (SD = 0.81) and 15.27 different DTs (SD = 2.01) are present on the same number of Suletice-Berand leaves. In total, 19 different DTs of five FFGs are present in the Seifhennersdorf assemblage, and 23 different DTs of six FFGs in Suletice-Berand.

Figure 6 Herbivory metrics compared between Seifhennersdorf and Suletice-Berand regarding whole assemblages and fossil-species phenology.

(A) Damage frequencies by Functional Feeding Groups (FFGs) compared between A1-A2: the whole assemblages, A5-A6: leaves of deciduous fossil-species, and A7-A8: leaves of evergreen fossil-species. Additionally, the proportions of leaves from deciduous and evergreen fossil-species in the assemblages are stated in A3-A4. (B) Damage type occurrences by FFGs compared between B1: the whole assemblages, B2: leaves of deciduous and evergreen fossil-species from Seifhennersdorf, and B3: leaves of deciduous and evergreen fossil-species from Suletice-Berand. The numbers above the columns in B1-B3 represent the total damage type occurrence. (C) Damage type richness compared between C1: both assemblages and C2: leaves from deciduous and evergreen fossil-species. The rarefaction curves are reduced to 400 leaves for visibility. For details regarding the metrics, see Material & Methods.

Figure 7 Insect damage types on Seifhennersdorf leaves.

(A) MMG PB Sf 4356 Carya fragiliformis with DTs 2, 5, and 14. (B) MMG PB Sf 5400 C. fragiliformis with DTs 14 and 81. (C) MMG PB Sf 1672 Carpinus grandis with DT 78. (D) MMG PB Sf 5159:1 C. grandis with DT 78. (E) MMG PB Sf 5181 C. grandis with DT 214. (F) Detail of K. Scale 0.5 cm. (G) MMG PB Sf 4812 C. fragiliformis with DT 16. (H) MMG PB Sf 5525 C. fragiliformis with DTs 50 and 57. (I) MMG PB Sf 1626 C. grandis with DT 214. (J) MMG PB Sf 931 C. fragiliformis with DTs 50 and 57. (K) MMG PB Sf 8360 Acer angustilobum with unknown gall DT. Unless otherwise stated, the scale is 1 cm.

Figure 8 Insect damage types on Suletice-Berand leaves.

(A) MMG PB SuBe 828a Sloanea olmediifolia with DT 20 (07). (B) MMG PB SuBe 25b Carpinus grandis with DT 78. (C) MMG PB SuBe 107c Platanus neptuni with DTs 18 and 38. (D) MMG PB SuBe 2:1c Engelhardia orsbergensis with DT 2. (E) Detail of A. Scale 0.5 cm. (F) MMG PB SuBe 911a Acer tricuspidatum with DT 2. (G) MMG PB SuBe 871t P. neptuni with DTs 5 and 12. (H) MMG PB SuBe 451a Acer palaeosaccharinum with unknown gall DT. The counterpart to I. Scale 0.5 cm. (I) MMG SuBe 634c A. palaeosaccharinum with unknown gall DT. Unless otherwise stated, the scale is 1 cm.

Most herbivory traces were caused by external foliage-feeding (Fig. 6). Common are hole feeding (DTs 2, 3, and 5), leaf margin feeding (DTs 12 and 14), and leaf skeletonization (DT 20). The assemblages differ regarding the richness of hole-feeding DTs (Seifhennersdorf: DTR400 leaves = 4.13 different DTs (SD = 1.14), Suletice-Berand: DTR400 leaves = 6.34 different DTs (SD = 1.21)) and leaf-skeletonization DTs (Seifhennersdorf: DTR400 leaves = 0.31 different DTs (SD = 0.50), Suletice-Berand: DTR400 leaves = 3.51 different DTs (SD = 0.96)). Comparable between the assemblages is the low frequency (DAM% < 0.5%), occurrence (DTO < 5 DTs), and richness (DTR400 leaves < 0.5 different DTs) of galling and leaf-mining DTs (Figs. 7F, 7K, 8C, 8H, and 8I), and the absence of DTs caused by piercing and sucking insects.

Insect herbivory regarding the TCT of leaves

Overall, DTs are most common on the abundant TCTs (n > 20 leaves, Fig. 3) but with intra- and inter-site differences (Fig. 9). First, TCTs differ regarding the damage frequency, especially in comparison between the assemblages. In Suletice-Berand, a TCT is more frequently affected than in Seifhennersdorf due to the overall higher incidence of insect herbivory. Within the assemblages, the differences in the damage frequency of TCTs are more pronounced in Suletice-Berand than in Seifhennersdorf. Damage frequencies vary between 2.85% (TCT B) and 5.89% (TCT F) in Seifhennersdorf and between 4.64% (TCT B) and 18.75% (TCT P) in Suletice-Berand (Fig. 9). Leaf types with toothed margins (TCTs E, F, and P) are more frequently damaged than those with entire margins (TCTs A and B). The same applies to leaf types with non-looped secondary venation (TCTs F and P, except TCT B) compared to those with looped secondary venation (TCTs A and E). Second, TCTs differ in the occurrence of DTs (Fig. 9). In both assemblages, DT occurrences tend to be higher on leaves with toothed margins (TCT E, F, and P) than on entire-margined leaves (TCTs A and B). In Seifhennersdorf, most DTs occur on unlobed and toothed leaves with non-looped secondary venation—TCT F. In Suletice-Berand, the number of DT occurrences is higher on leaves with looped secondary venation, especially on TCT E, but also on TCT A. Third, TCTs differ regarding DT richness. According to sample-based rarefaction, DT richness for a given TCT tends to be higher in Suletice-Berand than in Seifhennersdorf, except for TCT B (Fig. 9). In Seifhennersdorf, the DT richness is the highest on leaves of TCT F with in total 15 different DTs of three FFGs. For comparison, there are only five different DTs of two to three FFGs on TCTs A, B, E, and P leaves. Based on sample-based rarefaction, the DT richness in Suletice-Berand is also highest on leaves of TCT F, followed by TCTs E and A. In total, 15 different DTs are present on TCTs E and F, 11 on TCT A, and five on TCTs B and P.

Figure 9 Herbivory metrics compared between Seifhennersdorf and Suletice-Berand regarding Trait Combination Types (TCTs).

(A) Damage frequencies by Functional Feeding Groups (FFGs) compared between abundant TCTs in A1–A5: the Seifhennersdorf assemblage and A6–A10: the Suletice-Berand assemblage. (B) Damage type occurrences by FFGs compared between abundant TCTs in B1: the Seifhennersdorf assemblage and B2: the Suletice-Berand assemblage. (C) Damage type richness compared between C1–C5: abundant TCT in both assemblages. The rarefaction curves are reduced to 400 leaves for visibility. Abundant TCTs are represented with more than 20 leaves. For details regarding the metrics, see Material & Methods.

In both assemblages, hole-feeding DTs such as DTs 2–5 and leaf margin-feeding DTs such as DTs 12–14 frequently occur on leaves of different TCTs. Thus, they are not TCT-specific. There are only a few exceptions that indicate a relationship between the occurrence of certain DTs and TCTs. For example, TCT F leaves frequently showed traces of herbivory that were concentrated between secondary leaf veins (DT 78, Figs. 7C, 7D, and 8B) or at the junction of secondary and primary leaf veins (DT 50 and 57, Figs. 7H and 7J). Characteristic leaf skeletonization was observed on TCTs E and F in Suletice-Berand (DT 20, Figs. 8A and 8E). Rare evidence of leaf-mining was found on leaves of TCT E (Suletice-Berand, Fig. 8C) and of galling DTs on leaves of TCT P (Figs. 7F, 7K, 8H, and 8I).

Insect herbivory regarding fossil-species phenology and taxonomic affiliation

In the following, the relevance of the phenological and taxonomic background to the observed relationship between TCTs and insect herbivory is reviewed. The analyzed assemblages show distinct differences in the proportion of leaves originating from deciduous or evergreen fossil-species (Figs. 6A3 and 6A4). Also, herbivory metrics indicate meaningful intra- and inter-site differences regarding the relationship between insect herbivory and deciduous or evergreen leaves (Fig. 6).

The Seifhennersdorf assemblage contains a high proportion of leaves from various deciduous fossil-species, and with 76.73%, most of these leaves are assigned to TCT F (Table S4). Damage frequency, DT occurrence, and DT richness indicate an affinity of insect herbivores to fossil-species with these traits (Figs. 6 and 9). However, herbivory metrics also show a taxonomy-related influence on the observed relationship. Damage frequency varies between 2.61% (Ulmus fischeri) and 6.81% (Carya fragiliformis), DT occurrences between one (Salix varians) and 99 DTs (C. fragiliformis), and DT richness between one (S. varians) and 12 different DTs (C. fragiliformis). Fossil-species of Juglandaceae and Betulaceae differ remarkably from others by higher damage frequencies, DT occurrence, and DT richness (Table S4). Therefore, the DT record on leaves of Carya fragiliformis, Carpinus grandis, and Carpinus roscheri (Figs. 7A–7E and 7G–7J) is mainly responsible for the observed relationship between deciduousness, TCT F, and insect herbivory in Seifhennersdorf.

Unlike Seifhennersdorf, the Suletice-Berand assemblage is characterized by a higher proportion of leaves originating from evergreen fossil-species (Table S5). According to their frequencies, most of these leaves are of TCTs E (47.55%), A (27.04%), and F (7.17%). Leaves of deciduous fossil-species are mainly of TCTs E (33.23%), F (29%), and P (14.50%). Leaves of evergreen and deciduous fossil-species are more frequently damaged in Suletice-Berand than in Seifhennersdorf and show distinctly higher DT richness (Fig. 6). Within the Suletice-Berand assemblage, leaves of evergreen and deciduous fossil-species differ regarding DT occurrence and DT richness. Comparable to Seifhennersdorf, the evidence of insect herbivory in Suletice-Berand also varies between fossil-species with similar TCT or phenology—indicating the importance of taxon-specific traits for insect herbivory. The damage frequencies between the fossil-species varies between 5% (Conus studeri) and 28.57% (Acer tricuspidatum), the DT occurrences between one (C. studeri) and 49 DTs (Engelhardia orsbergensis), and the DT richness between one (C. studeri) and 11 different DTs (E. orsbergensis). Like in Seifhennersdorf, Juglandaceae leaves (E. orsbergensis) are most abundant in Suletice-Berand. Insect herbivores also frequently affected these leaves, accounting for a significant portion of the damage reported for TCT E (Fig. 8D and Table S5). TCT A leaves in Suletice-Berand also show stronger evidence of herbivory than in the Seifhennersdorf. That is related to frequently damaged evergreen Lauraceae leaves (Table S5). Leaves with non-looped secondaries (TCTs F and P) differ from leaves with looped secondaries (TCTs A and E) by showing higher damage frequencies and DT richness (Fig. 9). Again, significant portions of the DT record are related to individual fossil-species. For instance, the DTs on leaves of Sloanea artocarpites, an evergreen tree, and Zelkova zelkovifolia, a deciduous tree, significantly determined the herbivory metrics regarding TCT F (Fig. 9 and Table S5).

Multivariate analyses of insect herbivory

The different herbivory metrics could be partly explained with multivariate analysis (Dataset 5, Fig. 1). First, the presence/absence of DTs (interaction) varies significantly between the localities, among classes of TCTs (when secondary venation is not considered) and depends on the LMA (2). It confirms the abovementioned observations: traces are often found on leaves with relatively low LMA, from Suletice-Berand, and belonging to TCT class E/F. However, from this data (Dataset 5), no significant effect of TCTs is found when secondary venation is considered. Although included in the list of explanatory variables, leaf size does not explain the presence/absence of herbivory traces (2). Dataset 1, which does not provide quantitative trait measurements but enables to test of the effect of TCTs, phenology, fossil-species, and locality, reflects a relationship of herbivory (presence/absence) with the TCTs, especially TCT F (3). The models focusing on the presence or absence of DTs have extremely low R2, probably due to the low proportion of leaves showing DTs (i.e., ~10%, Table S6). That indicates that it is not possible to predict insect herbivory based on the quantitative traits and environmental/ecological variables used in the models.

Second, models analyzing the leaf surface damaged (area damaged and herbivory index in Dataset 5) reveal an important effect of phenology (as assessed with the modern relatives, models M4 and M5, Table S6). A positive correlation between the herbivorized leaf surface and the leaf size is also visible when the area damaged is the herbivory metric used (M4). Conversely, analyses made on the smaller Dataset 4 (without LMA imputation, Fig. 1) did not provide any information regarding DTs and their relationship to leaf traits and/or assemblage properties. The detailed results are provided in Article S2.

(2) Interaction~locality+LMA+classesofTCTs(+leafsize)

(3) Interaction~locality+TCTs

(4) Areadamaged~leafsize+phenology

(5) Herbivoryindex~phenology(+LMA)

Discussion

Leaf morphological-morphometric variability

TCT data help to evaluate the diversity and abundance of different leaf architectural types and, thus, the morphological variability of an assemblage of dicot angiosperm leaves (Roth-Nebelsick et al., 2017; Kunzmann et al., 2019). Belonging to the same vegetation type, namely the ecotone between broad-leaved evergreen and mixed mesophytic forest (Teodoridis & Kvaček, 2015), the assemblages of Seifhennersdorf and Suletice-Berand are expectedly similar concerning the TCT composition (Fig. 3). Differences between the assemblages are in the proportions of TCTs. Some traits in the TCT classification may be linked to the deciduous or evergreen habit and, therefore, leaf economics. For instance, leaf morphotypes with toothed margins and non-looped secondary venation types (e.g., TCTs F or P) tend to be more common in deciduous than evergreen species, whereas leaf types with entire margins and looped secondary venation types (e.g., TCT A) tend to be more common in evergreen species (Roth-Nebelsick et al., 2001; Walls, 2011; Royer et al., 2012; Li et al., 2016). The observed differences in the TCT proportions (Fig. 3) may therefore be consistent with local variations in the proportion of woody deciduous and woody evergreen angiosperms in the ecotonal early Oligocene vegetation, which is typically characterized by 50–60% of deciduous elements and 30–40% of evergreen elements (Teodoridis & Kvaček, 2015). This is supported by the taxonomical background of TCTs (taxonomy-based TCT approach, Table S2). Thus, the predominance of TCT F in the Seifhennersdorf assemblage (Fig. 3) is mainly related to diverse deciduous elements (Walther & Kvaček, 2007). The frequent occurrence of TCTs showing closed secondary venation types (e.g., TCTs A and E, Fig. 3) in Suletice-Berand is related to the greater abundance of evergreen fossil-species (Kvaček & Walther, 1995). The taxonomic diversity reflected by the different TCTs is not yet constrained. It varies from single fossil-species or genera (e.g., TCT P) to a higher taxonomic diversity (e.g., TCTs F and E). A better understanding of the relation between TCTs and taxonomy could be important for understanding the adaptive benefits of specific leaf architectures.

Quantitative leaf traits revealing a correlation between leaf size, LMA, and fossil-species further support that trait variations are partly based on the assemblage’s taxonomic composition (models M1 and M2). This shows that even if the identification of fossil floras is complex, it is necessary to reconstruct these traits at the scale of paleo-communities and by harmonizing the number of specimens per taxa (the abundancies of fossil plants in their environment at the time remains unknown and is avoided for the use of Community Weighted Means, Garnier et al., 2004). The quantitative analyses of the two assemblages highlight their strong resemblance, which confirms the previous descriptions of these floras (Kvaček & Walther, 1995; Walther & Kvaček, 2007). The occurrence of smaller leaves tending to higher LMA in Suletice-Berand and oppositely larger leaves with low LMA in Seifhennersdorf is consistent with (1) the higher portion of broad-leaved evergreen elements in the local vegetation of Suletice-Berand, as pointed out by Kvaček & Walther (1995), and (2) the dominance of broad-leaved deciduous and more azonal (riparian) elements in Seifhennersdorf, which grew on partly flooded and fertile soils (Walther & Kvaček, 2007). The absence of a significant correlation between TCTs and the parameters leaf size and LMA is likely explained by the taxonomic diversity contained in these categories (thus, TCTs are not entirely functional groups). In addition, intraspecific variability of leaf traits can be high and partially cloud interspecific trends (Siefert et al., 2015; Roth-Nebelsick et al., 2021). The poor number of replicates for some TCTs (e.g., TCTs C, D, O) may also have been an additional limit to determining possible relationships (especially since rare TCTs were excluded from multivariate analyses).

Insect herbivory pattern

In Seifhennersdorf and Suletice-Berand, DTs caused by external foliage-feeding insects were significantly more abundant than DTs of piercing and sucking, leaf-mining, or galling insects. Most of these external foliage-feeding DTs are characterized by low specificity and were found on multiple TCTs and unrelated host plants (e.g., DT 2, 3, 12, 14; Tables S4 and S5). The DTs were either caused by generalist insect herbivores feeding on multiple plant species, or the host plants were associated with various specialist insect herbivores that generated similar damages (Labandeira et al., 2007). It becomes difficult to distinguish between the external foliage-feeding DTs of generalists and specialists in the fossil record because various and even unrelated insects can cause comparable traces due to convergence in the evolution of mandibulate mouthparts in different insect orders (Carvalho et al., 2014; Labandeira, 2019). The taxonomic affiliation of external foliage-feeding DTs was currently found only in exceptional cases (e.g., Wilf et al., 2000; Wedmann, Wappler & Engel, 2009; Winkler et al., 2010; Adroit et al., 2020; Hazra et al., 2021). The skeletonization on leaves of Sloanea olmediifolia (Elaeocarpaceae) in Suletice-Berand (Figs. 8A and 8E) is comparable to a DT record of chrysomelids published by Adroit et al. (2020) on Parrotia persica (Hamamelidaceae) but also to external damages caused by adult curculionids (Boogs, 2018). No body fossils of leaf beetles were reported for the studied assemblages. However, most of the terrestrial beetles described from Seifhennersdorf belong to weevils (Tietz, Berner & Mätting, 1998; Prokop & Fikáček, 2007). Whether the same is true for Suletice-Berand to explain the DTs cannot be assessed due to the lack of description of the few remains of insects.

Leaf mines and galls are mainly related to specialized and host-specific insect herbivores. Their characteristic features allow a taxonomic determination by comparison with modern analogs (e.g., Labandeira, 2002, 2005; Knor et al., 2013; Hazra et al., 2020, 2021). The DTs 18 and 38 (Fig. 8C) recorded on leaves of Platanus neptuni (Platanaceae) from Suletice-Berand show such a specific pattern and were caused by leaf-mining larvae of Incurvaridae (aff. extant Paraclemensia, Labandeira, 1998, 2002). Both assemblages analyzed were characterized by the marked rarity of endophytic DTs. Comparable results have been reported from other Oligocene fossil sites, such as Enspel (Gunkel & Wappler, 2015). However, the reasons for that rareness still need to be made clear. As stated in Material & Methods, assessing the effect of chipping of coalified/compressed material on the occurrence and richness of galling or leaf-mining DTs is complicated. However, it could not be excluded as a possible reason.

Overall, the terrestrial insect body fossil record is sparse in both localities and hardly suggests a relationship to the DTs on leaves. However, remains of Cicadidae, which represent a group of plant sap-sucking insects, as well as of Bibionidae (Bibio) that feed on the nectar of flowering plants or honeydew of aphids, provide evidence of feeding strategies not recorded by DTs on leaves (Tietz, Berner & Mätting, 1998; Prokop & Fikáček, 2007). Consequently, insect body fossils and DT records may be complementary. In summary, the results of the present study allow no valid statements about significant differences in insect herbivore diversity between Seifhennersdorf and Suletice-Berand. Still, information on host-plant species affected by insect herbivory and their characteristic leaf traits can be derived from the data.

Insect herbivory and leaf traits

The data revealed different proportions of deciduous and evergreen elements in the Seifhennersdorf and Suletice-Berand assemblages. The nutritional quality and palatability of deciduous and evergreen leaves are essential criteria for the susceptibility of a host plant to be attacked by herbivorous insects (e.g., Coley & Barone, 1996; Wilf et al., 2001; Knepp et al., 2005; Royer et al., 2007; Pringle et al., 2011; Silva, Espírito-Santo & Morais, 2015). The nutritional quality of consumed leaves for optimal growth and survival rates of insect herbivores is determined by their ratio of carbon (C) to nitrogen (N) compounds. Needs vary between insect species, but the amount of nitrogen is generally vital (Schoonhoven, van Loon & Dicke, 2005). However, consuming N-rich photosynthetic plant tissues always goes with the uptake of C compounds from cellulose, lignin, cell wall hemicellulose, and secondary metabolites (Mattson, 1980; Scriber & Slansky, 1981; Schoonhoven, van Loon & Dicke, 2005). Depending on acquisitive or more conservative resource strategies, evergreen and deciduous species have different C/N ratios in their leaves. Evergreens follow a conservative resource strategy by having long-lasting, robust leaves with a high LMA (e.g., Givnish, 2002; Wright et al., 2004). To ensure leaf longevity, these plants need to invest a higher proportion of C compounds into structural tissues (non-photosynthetically active vascular and sclerenchymatous tissues) and secondary metabolites to make them more resistant against abiotic and biotic environmental impacts (e.g., Coley & Barone, 1996; Knepp et al., 2005; Pringle et al., 2011; Nakamura, Inari & Hiura, 2014; Silva, Espírito-Santo & Morais, 2015; de la Riva et al., 2016; Nascimento et al., 2019). Therefore, LMA and leaf life span (determined from modern relatives) are used as proxies for the C/N ratio of fossil-species based on correlations observed in present-day vegetation (e.g., Reich et al., 1991; Wright et al., 2004; de la Riva et al., 2016).

The frequency of large, toothed leaves with low mean LMA related to deciduous fossil-species in Seifhennersdorf (Fig. 5, Walther & Kvaček, 2007) suggests vegetation with leaves of low C/N ratio. Herbivory metrics (Figs. 6 and 9) observed within this assemblage reflect higher attractiveness of deciduous fossil-species with TCT F leaves to insect herbivores since deciduousness is also correlated with lower leaf thickness and robustness (Pringle et al., 2011; Silva, Espírito-Santo & Morais, 2015). Therefore, the leaf characteristics are instead not responsible for the low frequency of damaged leaves in Seifhennersdorf compared to those in Suletice-Berand (Fig. 6). Noteworthy is the absence of leaf-mining DTs in Seifhennersdorf, where favorable leaf properties appear (Figs. 4 and 5). In previous studies, leaf-mining richness was correlated with leaf traits, especially with low levels of sclerophylly, larger average leaf size, and longer leaf length (Sinclair & Hughes, 2008; Bairstow et al., 2010).

From its high proportion of evergreen leaves, one would expect lower evidence of insect herbivory in the Suletice-Berand assemblage. However, herbivory metrics indicate increased evidence of herbivory on evergreens compared to the Seifhennersdorf assemblage (Fig. 6). A possible explanation is that despite the lower palatability of evergreen leaves, herbivore damages accumulated over time on the leaves during their long lifespan (Southwood, Brown & Reader, 1986). In addition, the probability of fossilizing highly damaged deciduous leaves was likely lower. It is also conceivable that increased damages on evergreen hosts may be related to co-occurrence with Fabaceae. Fabaceae have been attributed to be beneficial for ecosystem productivity due to their ability to fixate atmospheric nitrogen through symbiosis with microorganisms in the rhizosphere (Matos et al., 2021). Currano et al. (2016) hypothesized a possibly positive buffer effect on leaf nutritional quality by Fabaceae in ecosystems under elevated pCO2 conditions, which are expected to reduce leaf nutritional quality (Stiling & Cornelissen, 2007). Legumes in the forest ecosystem may have facilitated higher leaf nitrogen concentration across fossil-species through the enrichment of nitrogen in the soil and thus positively affect the leaf nutritional quality of other plants, even evergreens (Matos et al., 2021). In addition, the presence of Fabaceae leaves in the vegetation may also have affected insect behavior. By having these nitrogen-rich leaves in their environment, insects could be less selective on the quality of leaves (e.g., of evergreens) to meet their physiological requirements (“dietary mixing”). However, the number of DTs on legume leaves in the data is too low to support this hypothesis for the present study.

Moreover, insect herbivory was more common in some TCTs than in others. Insect herbivory occurred mainly on toothed leaf morphotypes, especially TCT F and TCTs E and P. However, leaf morphology or leaves’ TCTs cannot sufficiently explain DTs’ frequency, occurrence, or richness. It is a more complex relationship, where—besides leaf morphological characteristics or TCT—LMA, phenology, and taxonomic affiliation are important. In both assemblages, evidence of insect herbivory was mainly based on frequent fossil-species and their traits (e.g., Betulaceae, Juglandaceae, Elaeocarpaceae, Platanaceae, or Sapindaceae). Therefore, observations of trends based on higher-ranked categories like assemblage, TCTs, or phenology should be viewed cautiously and cannot be considered generally valid. There are potentially other host plant-specific traits not investigated here that also shape the interrelationship with insect herbivores, for instance, micromorphological traits or plant chemical composition. The latter will be difficult to estimate from fossil-species. Extensive studies of modern relatives could be an approach. In addition, more focused studies on the DT record of specific fossil-species (e.g., documenting their DTs from different fossil sites) could help to evaluate the DT data of a fossil-species observed in a new assemblage. Based on such studies, assessing the presence, absence, and richness of DTs or FFGs on fossil-species will be more profound. Such investigations still needed to be done for the fossil-species in our study.

Biases in the fossil leaf and DT record

Discussion on potential biases in the fossil leaf and DT record on the methods used can be found in Article S3.

Conclusions

1) Despite differences in leaf size and LMA, the fossil floras of Seifhennersdorf and Suletice-Berand share similarities regarding their quantitative traits, reflecting close temporal, paleoclimatic, paleogeographic, and depositional contexts. The floristic compositions, especially the proportion of broad-leaved deciduous and broad-leaved evergreen elements, are reflected in TCT frequencies,

2) Multivariate analyses provided more insights into leaf size and LMA variations, sometimes yielding different results than analyses with a single explanatory variable. This shows the importance of using more advanced approaches, such as GLMs or PCA, to understand trait variation in (and between) fossil assemblages. The imputation of LMA data allowed more specimens to be included in the analyses and seems promising, given the low preservation frequency of petioles. Selecting leaves for quantitative leaf trait calculations requires specific attention regarding the characteristics of the individual assemblages (e.g., determination of fossil-species, number of specimens, or preservation state).

3) In Seifhennersdorf, DTs’ frequency, occurrence, and richness are highest on deciduous fossil-species. Because of the potentially more favorable leaf characteristics, the low incidence of leaf damage compared to Suletice-Berand cannot be explained. The results conflict with the Suletice-Berand assemblage, consisting of evergreen leaves potentially less favorable to herbivorous insects, having a higher proportion of differently damaged leaves.

4) An association between insect herbivory and toothed leaf morphotypes, or TCTs (E, F, or P), was observed and suggested that these TCTs may be associated with host plant species with more favorable traits.

5) The taxonomic composition significantly influenced trait variations (e.g., leaf area and LMA) and the evidence of insect herbivory. Insect herbivory was most pronounced on leaves of abundant fossil-species (e.g., of Betulaceae, Juglandaceae, Elaeocarpaceae, Platanaceae, or Sapindaceae).

Supplemental Information

Supplemental Information 1 Geological background and collection history.

Additional information regarding the geological background, fossil site features, and collection history.

Click here for additional data file.

Supplemental Information 2 Results of the multivariate analyses without imputation of LMA.

The results of multivariate analyses performed on Dataset 4 without LMA imputation (n = 228 specimens; see Material & Methods) are shortly presented.

Click here for additional data file.

Supplemental Information 3 Biases in the fossil leaf and insect damage type record.

Additional discussion regarding the effect of potential biases in the fossil leaf and insect damage type (DT) record on the methods used.

Click here for additional data file.

Supplemental Information 4 Herbivory data matrix for sample-based rarefaction (fossil-species Seifhennersdorf).

Matrix (Seifhennersdorf) used for rarefaction procedures to calculate damage type richness for the whole leaf assemblage and for fossil-species with at least 20 leaves. X1-X280 refers to the damage type numbers in Labandeira et al. (2007).

Click here for additional data file.

Supplemental Information 5 Herbivory data matrix for sample-based rarefaction (phenology of fossil-species Seifhennersdorf).

Matrix (Seifhennersdorf) used for rarefaction procedures to calculate damage type richness related to phenology (evergreen vs. deciduous) of fossil-species. X1-X280 refers to the DT numbers in Labandeira et al. (2007).

Click here for additional data file.

Supplemental Information 6 Herbivory data matrix for sample-based rarefaction (TCT Seifhennersdorf).

Matrix (Seifhennersdorf) used for rarefaction procedures to calculate damage type richness related to Trait Combination Types (TCT) of fossil-species. X1-X280 refers to the DT numbers in Labandeira et al. (2007).

Click here for additional data file.

Supplemental Information 7 Herbivory data matrix for sample-based rarefaction (fossil-species Suletice-Berand).

Matrix (Suletice-Berand) used for rarefaction procedures to calculate damage type richness for the whole leaf assemblage and for fossil-species with at least 20 leaves. X1-X280 refers to the DT numbers in Labandeira et al. (2007).

Click here for additional data file.

Supplemental Information 8 Herbivory data matrix for sample-based rarefaction (Phenology of fossil-species Suletice-Berand).

Matrix (Suletice-Berand) used for rarefaction procedures to calculate damage type richness related to phenology (evergreen vs. deciduous) of fossil-species. X1-X280 refers to the DT numbers in Labandeira et al. (2007).

Click here for additional data file.

Supplemental Information 9 Herbivory data matrix for sample-based rarefaction (TCT Suletice-Berand).

Matrix (Suletice-Berand) used for rarefaction procedures to calculate damage type richness related to Trait Combination Types (TCT) of fossil-species. X1-X280 refers to the DT numbers in Labandeira et al. (2007).

Click here for additional data file.

Supplemental Information 10 R-code for multivariate statistical analysis.

See Material & Methods regarding the multivariate statistical analysis applied.

Click here for additional data file.

Supplemental Information 11 Analysis of the leaf mass per area (LMA) imputation process.

There are two types of LMA: calculated LMA shown in blue and predicted (simulated) LMA in red. Each imputed value (data point) averages 10 imputation cycles per specimen. (A) The relationship between LMA and leaf area is shown for both types of LMA. Above are the associated density curves. (B) Compared are both types of LMA per locality. (C) Compared are both types of LMA per fossil-species. Five simulated LMA values are below the range of the calculated LMA. They were removed from the multivariate analyses. See Material & Methods for further information.

Click here for additional data file.

Supplemental Information 12 List of all analyzed specimens and accession numbers.

Click here for additional data file.

Supplemental Information 13 Qualitative leaf data.

All taxonomic, Trait Combination Type, and insect damage type determinations as well as quantitative leaf trait measurements (preservation index, leaf area, leaf mass per area, leaf length, leaf width, length/width ratio) for all studied leaves or graphical digitized leaves.

Click here for additional data file.

Supplemental Information 14 Quantitative leaf data for multivariate analysis.

All taxonomic, Trait Combination Type, and insect damage type determinations as well as quantitative leaf trait measurements (preservation index, leaf area, leaf mass per area, leaf length, leaf width, length/width ratio) for all studied leaves or graphical digitized leaves.

Click here for additional data file.

Supplemental Information 15 Mean leaf mass per area (LMA), phenology, taxonomy-based TCT, and herbivory data for taxonomic units with at least 20 leaves in the Seifhennersdorf assemblage.

Click here for additional data file.

Supplemental Information 16 Mean leaf mass per area (LMA), phenology, taxonomy-based TCT, and herbivory data for taxonomic units with at least 20 leaves in the Suletice-Berand assemblage.

Click here for additional data file.

Supplemental Information 17 Results of Type-II ANOVA.

‘ANOVAs were calculated on GLMs, including calculated and simulated LMA (Dataset 5, n = 506 leaves). Chi-square values are displayed only for at least marginally significant variables, along with p-values. Some variables were excluded from the modeling (sometimes to allow the model to converge) and are marked as “not tested”’.

Click here for additional data file.

The authors thank the scientific editor Peter Wilf, the reviewer Dana Royer, and two anonymous reviewers very much for processing the manuscript and the extensive scientific reviews.

The authors cordially thank Dimitra Mantzouka (Athens, Greece), Jörg Schneider (Freiberg, Germany), and Pierre Ganault (Leipzig, Germany) for their support and helpful discussions during the project. Many thanks go to the employees of the paleobotanical section of the Senckenberg Natural History Collections Dresden for supporting the authors with digitizing the fossil leaves. Lutz Kunzmann thanks the late Zlatko Kvaček and the late Harald Walther for former scientific discussions on the studied fossil floras.

Additional Information and Declarations

Competing Interests

Author Contributions

Data Availability

The authors declare that they have no competing interests.

Christian Müller conceived and designed the experiments, performed the experiments, analyzed the data, prepared figures and/or tables, authored or reviewed drafts of the article, and approved the final draft.

Agathe Toumoulin analyzed the data, prepared figures and/or tables, authored or reviewed drafts of the article, and approved the final draft.

Helen Böttcher performed the experiments, prepared figures and/or tables, and approved the final draft.

Anita Roth-Nebelsick conceived and designed the experiments, authored or reviewed drafts of the article, and approved the final draft.

Torsten Wappler conceived and designed the experiments, authored or reviewed drafts of the article, and approved the final draft.

Lutz Kunzmann conceived and designed the experiments, performed the experiments, authored or reviewed drafts of the article, and approved the final draft.

The following information was supplied regarding data availability:

The raw data are available in the Supplemental Files.

All analyzed specimens are stored in the Senckenberg Natural History Collections Dresden, Museum of Mineralogy and Geology.

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
