# Peer review of "An integrated leaf trait analysis of two Paleogene leaf floras"

_PeerJ, doi:10.7717/peerj.15140_

## Round 0.1 · original submission · Major Revisions

· Academic Editor

Major Revisions

Thank you for submitting your work to PeerJ. We are fortunate to have received three extremely helpful reviews from specialists in the area of your manuscript, two of whom provided additional constructive comments in annotated pdfs. I agree with essentially all the suggestions made. All the reviews were positive about the general goals of the paper and the large dataset that you have carefully put together. Reviewer 3, in particular, felt that the paper still needs a big lift to get to the level needed for the journal and its readers, and I agree. However, that reviewer was highly encouraging and constructive, providing you with an exceptional number of helpful and specific suggestions to help you get there.

Overall, there is a consensus that your work is very worthwhile, but the paper would indeed benefit from some more effort toward sharper organization, more conciseness, and clearer focus and reader motivation when stating its goals, methods, and main conclusions. Therefore, my decision is Major Revisions.

Below is a quick summary of the main points made by one or more reviewers each, with a few from me added. Please consider and respond to these and all other points raised (many specific suggestions are found in the reviews). I look forward to your revision.

—General enthusiasm was expressed by all for your approach of combining leaf-trait and herbivory analyses in fossil leaf floras and recognition that this represents important innovation in the field, which has taken a simplistic view of the issue.

—The motivation for the study and the ILTA method needs to be much more clear, especially in the “goals” paragraph at the end of the Introduction. Please carefully phrase what major question you are asking and why it is important and of broad interest (see specific suggestions in reviews). Some of the text that currently leads the Discussion would go better in the goals paragraph.

—Please explain why these particular fossil sites were selected to study. Also, one site appears to have much better preservation than the other. Please add text addressing this issue and how it could affect the results, such as the rarefactions. Finally, you have a table with basic geological information about the sites, but it doesn’t seem to be cited where needed around line 206, please add citation if needed.

—How were evergreen and deciduous species distinguished?

—If you developed new R code, please make it available so others can use it.

—Please explain why you chose PCA as the ordination method.

—Please make sure that goals and approaches stated in the Introduction and Methods are executed. If not, remove them. Please remove redundant or unrealized analyses and metrics (i.e., choose one of the three methods for measuring leaf size and eliminate the three sets of definitions taking up lines 305-327).

—The TCTs are very interesting, please develop those explanations and analyses for the reader (reviewer 3 has many suggestions). In the figures, some graphical representation of the TCTs, perhaps as small leaf icons, would be very helpful, since very few readers are going to look them up.

—The Discussion needs a thorough revision. Paragraphs are long and hard to follow, please break them up into concise paragraphs that make more focused points. Conclusions (and the end of the Abstract) should state clearly what the major findings of your work are and why they are of broad interest, rather than state a long series of uncertainties.

—Please explain how you are using “zonal” and “azonal” here. The meaning seems different from the usage in ecological literature.

Moving here into some additional comments from me:

—Lead paragraph of Introduction seems mostly unnecessary as general statements about the fossil record that don’t relate to your study directly. You could probably just delete it. Text starting at line 102 is not entirely correct, only listing one of many reasons plants can be preserved without animals. Siliciclastic lacustrine deposits can preserve animals and plants together extremely well, for example.

—Line 142: “We will also assume…” Please clarify this sentence and make sure there is follow- up on it later in the paper (or delete it).

—Line 152: “What is a “volcanic-shaped mountain range”? Please reword.

—Line 202, citation of Koch (2011): Please consider carefully whether Koch should be an author here, given the large amount of data in their thesis that you revised here and the fact that Koch never got publication credit for their work.

Figures:
—Several graphs need some polish added in Adobe Illustrator or similar. Please eliminate background grids from graphs. Follow journal guidelines for illustrations.
—Several graphs are unnecessarily large for the amount of information presented. Please shrink and combine similar figures into multipart panel figures and be economical of page space.
—Colors chosen to represent the two sites in the graphs are too similar for red-green color blind readers (10% of your male audience, including me). Suggest black, white, gray, and no more than one each of colors in or near red, blue, and yellow.
—Please brighten and add contrast to some of your fossil photos.
—As mentioned above, some sort of visual icon set to help reader visualize the TCTs would be a terrific addition to the TCT figures. Otherwise they are “just numbers” to the reader.

—Tables: there are a large number of Tables. Please consider consolidating similar tables, moving some to supplementals, and otherwise saving page space, which leads to a better reader experience.

—Citations and references: these are not formatted for the journal. Please carefully follow journal formatting for in-text citations and references, referring to recent articles in the journal as examples. Not required, but adding hyperlinks to your DOIs is a big help to the production editors.

—General writing and organization will need some careful attention. The reviewers made many specific suggestions that will be helpful.

Again, we look forward to your revision!

·

Basic reporting

Quite a lot of awkward syntax and grammar. I only note a few examples in my general comments below.

Experimental design

Well done.

Validity of the findings

Well done.

Additional comments

Muller and colleagues present information on insect damage, floral taxonomy, leaf architecture, and leaf ecology for two Oligocene fossil floras. They then analyze these multiple dimensions in a technique they call integrated leaf trait analysis. This kind of multidimensional framework is the direction the paleobotanical community is moving, and the authors do well in providing an explicit framework. This paper will be a useful contribution for many paleobotanists.

Overall, I think the manuscript is in good shape. I start with two suggestions, followed by a list of more minor comments.

1) The motivation for Integrated Leaf Trait Analysis should be set up better (final paragraph of Introduction). There are four key elements: taxonomic composition of site, insect damage, leaf architecture, and leaf ecophysiology (leaf lifespan and leaf mass per area). A broader context of possible linkages between these elements is not presented, but should be. For example, why do the authors think that certain leaf shapes will suffer more insect damage (lines 139-141)? Or, why might insect damage be linked to the leaf economics spectrum (e.g., Wilf et al. 2001 and Royer et al. 2007)? I see that the authors explain many of the linkages in the Discussion. This background (the scaffolding) needs to be added to the Introduction.

2) For each site, cross-plots of LMA vs % damage and LMA vs DT richness should be made (from Tables 2-3). This will aid in the arguments made in the Discussion sub-section “Relation between insect herbivory and leaf traits” (see, for example, Figure 4 in Royer et al. 2007).



Smaller stuff:

Line 78: remove “are” (duplicated)

Line 101: “biased, challenging…”

Line 129: “Royer et al.”

Lines 131-132: You start this sentence with “However”, but it’s not clear how this sentence is linked to the preceding sentences about ecophysiological traits.

Line 132: What do you mean by “disparity”? Differences between species? Sites?

Line 161: need comma after “plains”.

Line 219: Need to define MMG

Line 435: To be clear, only the insertion point of the petiole into the blade needs to be preserved, not the entire petiole (you mention this later, but not here).

Lines 489 & 492: The two decimal places are not needed here (overly precise)

Line 537: Is 1.23 +/- 1.06 truly different than 0.78 +/- 0.72? Is this based on the Mann-Whitney test?

Line 540: I see from the Methods that the cutoff of 20 was chosen because every species has at least 20 specimens. But when comparing deciduous to evergreen leaf habits (Figure 1B), it looks like you can rarefy to ~300 specimens at SuBe and ~550 specimens at SF, and that the deciduous vs. evergreen differences are probably significant at both sites. Why did you rarefy to 20 specimens here?

Line 659: “over time”

Line 705: Is it possible with these historical collections that only the “good” specimens were retained (that is, lacking insect damage)? The value of insect-damaged fossil leaves was not fully appreciated at the time that these fossils were collected. Most modern collections are made in a more unbiased manner.

Lines 817 & 821: “leaf life span”

Table 1: CLAMP should not have the superscripted note #4.

Tables 4 & 5 captions: “400 *leaf* sample…”

Supplement: Some sort of guide is needed to define all of the abbreviated column headers in Documents S1-S6 and Tables S1-S3.

Reviewer 2 ·

Basic reporting

I made some minor grammatical suggestions in the attached word document, but overall the English is understandable. Sufficient background and references are provided, and the paper follows a standard structure.

Experimental design

The paper introduces “Integrated Leaf Trait Analysis”, which combines analyses of insect damage types, leaf area, leaf mass per area, and trait combination types. The experimental design is appropriate for the questions posed.

Line 150 - Fossil leaf collections from Seifhennersdorf and Suletice-Berand, two early Oligocene fossil localities, are used as case studies for Integrated Leaf Trait Analysis. Were there any particular reasons why these sites were chosen for the study? There are many comparisons made between the sites throughout the paper, so if the sites were chosen for reasons other than/in addition to their function as test cases for ILTA, that could be mentioned here.

Line 249 - What methods were used to categorize species as deciduous vs. evergreen (LMA, leaf shape?). Are these based on assignments from the papers cited in Supplemental Table 3? I suggest explaining this in the Methods.

Line 329 – Thank you for including the DT and leaf trait data in the supplement. I suggest added a supplemental file with any new R code used in this study for reproducibility.

Line 336 – Why was PCA used instead of other ordination methods (NMDS for example)? I suggest justifying this choice here.

Validity of the findings

Overall, the data is robust and the authors explain their results and interpretations in detail through the text and figures.

362 “The areal extent of herbivory pattern on the leaf blades is conspicuously low” – Is this observation based on the percentage of damaged leaves? I think this is difficult to say without surface area measurements of the insect damage and comparisons with other floras.

Line 562, 709-710 – “generalist insect herbivores” – Generalized damage types defined by Labandeira et al., 2007 are not exclusively made by insects that feed on a variety of hosts. For example, DT2 was found on multiple plant species in this study. This could mean that one or more insect species made DT2 damage on multiple plant species, plant species were associated with different specialized insect species that made similar DT2 damage, or a combination of these two scenarios occurred. Because a variety of mandibulate insects are known to make similar external foliage feeding damage, telling the difference between external foliage feeding made by generalists and specialists can often be difficult or impossible.

Lines 573-576, 828 – The authors suggest that the DT20 skeletonization damage on Sloanea olmediifolia was made by leaf beetles (Chrysomelidae). I agree that chryomelids are likely culprits, but there are other possibilities. Here’s an example of a coccinellid that makes elongate skeletonization/surface feeding zones: https://nzacfactsheets.landcareresearch.co.nz/factsheet/InterestingInsects/Hadda-beetle---Epilachna-vigintioctopunctata.html
Also, as mentioned in the text, curculionid fossils have been reported from Sf, and their larvae are typically woodborers. However, some adult curculionids feed on leaves - see the fourth picture down at this link for an example of surface feeding: https://bygl.osu.edu/node/1103).

Additional comments

I enjoyed reading this paper, and I’m interested in seeing the ILTA methods applied to more floras in the future.

Annotated reviews are not available for download in order to protect the identity of reviewers who chose to remain anonymous.

Reviewer 3 ·

Excellent Review

This review has been rated excellent by staff (in the top 15% of reviews)
EDITOR COMMENT
This is a model review for anyone wondering how best to tell authors, especially student authors, supportively that major changes are needed. The reviewer went to the mat to help the authors by making large numbers of very specific suggestions, all while maintaining a very positive and encouraging tone.

Basic reporting

See additional comments.

Experimental design

See additional comments

Validity of the findings

See additional comments

Additional comments

I was really excited to receive this paper to review because of the integration of leaf traits and herbivory data. It has become pretty standard in our field to look for relationships between leaf mass per area and herbivory, but very few if any studies have tried to look at leaf traits beyond that. I commend the authors for pushing the envelope! Thank you also for providing the full dataset in a format that can be easily analyzed by others.

I have recommended major revisions because I think a lot of work is needed to reach PeerJ standards. My concerns are summarized below, and also in the marked up pdf.

1. The introduction states that the primary purpose of the manuscript is to: “analyze and compare the richness and frequency of insect damage and its relationship to taxonomic composition and morphological-morphometric leaf traits of two fossil leaf assemblages from the European Paleogene”. This is a great idea, and novel! However, I felt that the manuscript fell short of actually doing this. The discussion of herbivory vs. leaf lifespan is sufficient, but more extensive analyses are necessary to document other relationships that exist (or do not exist, which is also a useful result). Here are some pointers to help achieve your goal:
a. Herbivory metrics for each taxon at each site are reported in the tables, but there is not enough written of how these metrics vary. Flesh out the first two sections of the results with an entire paragraph about how herbivory varies among taxa at each site.
b. I’d also like to see more in-depth analysis of herbivory variations among the TCTs- this is briefly touched upon in lines 534-538 and 542-546, but I’d like to see damage frequencies, richness, and FFG distributions for each TCT at each site. You might find some interesting results- for instance, do teeth affects the frequency of margin feeding. And then it would be useful to discuss whether there are differences between the two sites in the herbivory observed on the TCTs.
c. Are there any relationships between herbivory and the various measurements you made?
d. Could you use GLMs to discover what drives differences in herbivory among taxa (similar to what you did for the leaf traits, and which gave super cool results there)? You could use taxonomy, locality, preservation index, and leaf traits as predictors for different herbivory metrics and see if any of the models are significant.

2. Another primary goal of the manuscript is to “attempt to find out if special leaf architectural types (e.g., toothed leaves with pinnate-brochidodromous venation or toothed leaves with pinnate-craspedodromous venation) might be preferred by herbivorous insects and – if so – whether there is a further link to leaf longevity or not.” I have a couple recommendations to help better meet this objective.
a. You need to frame this goal better throughout the manuscript. In the introduction, please add a paragraph explaining why you might expect to find differences in herbivory among architectural types.
b. Rather than lumping together traits in the TCTs, would it make more sense to consider traits individually? You could first compare traits individually, and then build GLMs to look at traits in concert.

3. I found the manuscript rather hard to follow because of the sheer number of analyses used, including multiple variations of some analyses. I highly recommend that the authors streamline the manuscript to better highlight key results and address their stated goals. Here are two examples where I would recommend cutting, and I encourage the authors to find more.
a. Why do you need three different methods to quantify leaf size (actual measurements, Webb’s size classes, and Wolfe’s size classes)? All show larger leaf size at Seifhennersdorf, so just pick one for the main manuscript. I really like the analyses of the actual size measurements, and it integrates well with the multivariate analyses, so my recommendation would be to eliminate the size class data.
b. A large number of herbivory metrics are discussed in the methods, but then many of them are rarely discussed after that (DTOs, FFG metrics). It would be easier on the reader if fewer metrics reported are reported and there is detailed discussion of each metric.

4. Additions are needed in the methods section. A reader should be able to understand what you did with only minimal reference to other papers. Please add the following to your manuscript:
a. How much stratigraphic thickness, and, if possible time, is thought to be represented by each fossil assemblage?
b. How was leaf life span was determined for each taxon. Was this using nearest living relatives, or the leaf mass per area values? If the former, how confident are you that leaf lifespan could be correctly designated? Are there analyses that show how much this trait is conserved across species within each genus? If the latter, what cut-off did you choose and why? Royer et al. 2010 used 129 g/m2 as the cut-off, but if you look at the data in their Figure 2, you can see that there is a lot of evergreen taxa have LMA below this, and there are also a significant number of deciduous taxa above 129g/m2.
c. What exactly is PI and how was it determined?
d. Was there any preservation cut-off for the leaves to be included in the herbivory analysis? e.g., tertiary venation visible or at least half of a leaf?
e. Readers may not be familiar what the different TCTs are, and they might not have access to the Roth-Nebelsick paper in which they are described. Please either reproduce Figure 2 of Roth-Nebelsick et al. 2017, or include a figure that illustrates the abundant TCTs in your study.
f. I had trouble understanding how exactly the three methods of determining TCT were used in conjunction to categorize specimens. For example, say that Taxon A has some toothed leaves and some untoothed leaves. Did you include all leaves of that taxon in both TCTs, or did you categorize individual specimens based on their characters? If you are looking for correlations between herbivory and TCTs or between LMA and TCT, then I think you would want to do the latter. Similarly, if you know that a particular taxon could be placed in more than one TCT, and you did not have the appropriate features to determine the TCT on a particular specimen, should you have used that specimen in the analysis?

5. The manuscript would benefit from some reorganization and improvements to the grammar, syntax, and word choices.
a. I thought the first sentence of the discussion framed the paper really nicely: “The aim for applying ILTA to leaf assemblages is to characterise abundances and properties of dicotyledonous angiosperm leaf morphotypes in a palaeovegetation and thus to make conclusions about characteristics of the potential insect herbivore diet.” Following this, I would recommend that in the methods, results, and discussion, you first discuss the plant traits and then discuss the herbivory.
b. I’ve attached an annotated pdf that I hope will help you improve the clarity and the writing. I made some specific corrections and highlighted in yellow sentences/phrases/words that need fixing. I likely missed some.

6. The discussion section as a whole needs a massive re-write, and new points might become important if additional analyses are added as discussed above. Here are some specific pointers for improving the discussion:
a. Clearly separate discussion of intra-site variation from inter-site variations. This could help clarify what is driving observed patterns and how differences among taxa within a site could scale up to produce differences among sites.
b. Reorganize and add more to the “Trait Combination Types and quantitative leaf traits” section in the discussion. I’m not clear on what the main points you are trying to make are, and this section would benefit from additional discussion on what we know about leaf traits in modern environments, such as the relationship between leaf size and precipitation (and if you buy the CLAMP results in Table 1, then SuBe is drier) or the observation that canopy leaves tend to have higher LMA than understory leaves. Another point that is important to address here, or if you choose later in the biases subsection, is the possibility that taphonomic biases could lead to different leaf sizes, LMA, and TCTs at the different sites. Slightly higher energy conditions and greater transport would result in an assemblage with smaller, lower LMA leaves. How can you rule this out? Or can you?
c. There is no discussion of other factors beyond the leaf traits that you measured which could be affecting herbivory metrics. What are possible mechanistic reasons why margin type, lobation, or venation type affect herbivory? Chemical defenses deserve mention, and given that the taxonomy of the sites is so well known, could you use any information from nearest living relatives to infer chemical defenses?
d. Herbivory is noted to be low at Sf compared to other Oligocene sites in Europe. Based on the photographs shown in Figure 2 & 3, it appears that preservation at Seifhennersdorf is not as good as at Suletice-Berand and perhaps at other European sites. Please address this in the discussion and explain whether this could play a role in the differences in herbivory observed.

Annotated reviews are not available for download in order to protect the identity of reviewers who chose to remain anonymous.

---

## Round 0.2 · Major Revisions

· Academic Editor

Major Revisions

Dear Christian and colleagues,

Thank you for your extensive revisions in response to the three expert reviewers and myself. The paper is considerably improved. The same three reviewers have now kindly checked your revision, and all agree that the paper is interesting to the community and much closer to publication-ready, or that it could be published as-is. The reviewers provided a list of additional helpful suggestions and markup files, to which I add below. Nearly all suggestions are minor in nature, but cumulatively they could take some time to implement carefully (though much less than the first round), and thus my decision is “major revision.” However, I do not intend to send the manuscript out for review again. I look forward to your revision!

Editor’s suggestions:
—Despite your rebuttal statements, it looks pretty obvious (comparing Figs. 8 and 9) that preservation of leaves and damage is considerably better at Suletice-Berand. I recommend being open about this from the start, rather than trying to convince us that it is not so (or being much more convincing), and including some text about what it may mean for the analyses. This may require some effort in parts of the text but will result in many fewer problems for you later.

—There are still a fair number of ungrammatical sentences. Please keep in mind that PeerJ does not copy-edit articles. This saves you a lot on the APC charge, but it also makes perfect writing the authors’ responsibility. In addition to the reviewers’ helpful markups, I strongly suggest using next-generation grammar software such as Grammarly or PaperPal to help you finish the paper (I use both routinely, and they also help to lighten wordy sentences). I will not be able to accept the paper until the copy is clean and ready to publish.

—Text is overall fairly wordy. I urge you to be critical to determine what you really need to say and cut the length by at least about 15%. Shorter papers are read more often and better understood.

—Please consolidate figures with similar information into single multipanel figures to save space (Figs. 2 and 3 => Figs. 2A,B).

—Please move some of the tables (at least Table 3, perhaps Table 2) to supplemental tables, where they are easily accessed online.

—Please indent the first lines of paragraphs (except for the first paragraph in a section).

—Please use hanging indent in the bibliography and embed the DOI hyperlinks (see PeerJ articles for examples). This will save a lot of processing time.

—There are still many very-long paragraphs. Please break these up into smaller paragraphs for readability.

—Title is a bit oddly phrased. Try something more straightforward, like “Integrated leaf-trait analysis of two Oligocene floras and their insect damage”?
—Abstract is very acronym heavy (LMA, TCTa-p, DT, ILTA) and thus not very readable. Please reduce acronyms in the abstract wherever possible for readability. “TCT” needs to be defined at first use.
—Line 303: header “leaf manual reconstruction” is ambiguous (i.e., is “leaf” or “maunal” being reconstructed?). How about “Reconstructing leaf shapes”?
—Line 350: “inference on” => “inference of” or “inferring”
Also in this section (and elsewhere) I suggest acknowldging that these LMA categories provide a useful approximation of phenology, but this is not an exact correlation as some readers might infer.

—Line 360 and below: “Firstly” => “First”, “Secondly” => “Second”

—Fig. 5 caption: “imputation” means “accusation”. Choose another word.

—Fig. 7: please shorten the long caption and do not use captions to explain Methods. You can save caption text by stating “see Methods for details.”

·

Basic reporting

Fine

Experimental design

Strong

Validity of the findings

Fine

Additional comments

The authors have done an excellent job incorporating feedback from me and the other reviewers. I think it is ready for publication.

I want to point out that in line 526, your generalized linear model for LMA includes leaf area as an explanatory variable. This strikes me as not ideal, seeing as leaf area is part of the LMA definition. This may violate some assumptions needed for GLMs.

Reviewer 2 ·

Basic reporting

The grammar and syntax has been improved. I attached a document with some minor suggestions.

Experimental design

My major questions from the first round of reviews were satisfactorily addressed. The authors included explanations of why the two sites (Seifhennersdorf and Suletice-Berand) were chosen and included more information on the methods used to categorize deciduous vs evergreen leaves. The authors also added their new R code as a supplementary file.

Validity of the findings

The manuscript (including figures) has been extensively revised, and I found the updated version of the results and discussion easier to follow.

Additional comments

Nice work!

Annotated reviews are not available for download in order to protect the identity of reviewers who chose to remain anonymous.

Reviewer 3 ·

Basic reporting

See additional comments

Experimental design

see additional comments

Validity of the findings

see additional comments

Additional comments

I was Reviewer 3 on the initial submission of this manuscript to PeerJ, and I am delighted at how much improvement has occurred! Thank you, authors, for taking the review comments to heart and doing such an extensive revision. I found this manuscript much easier to follow and a very nice first step at integrating leaf characters and insect herbivory. I encourage the authors to continue research along these lines and to consider doing future work that investigate mechanistic reasons why different TCTs are susceptible to different amounts of herbivory, acknowledging that this is beyond the scope of the current contribution.*

*I found this paper fascinating and has lots of food for thought, shall we say: Vermeij, G.J., 2015. Plants that lead: do some surface features direct enemy traffic on leaves and stems?. Biological Journal of the Linnean Society, 116(2), pp.288-294. https://academic.oup.com/biolinnean/article/116/2/288/2440324

Below are my suggestions for revisions, with more substantial comments first and then very small line-by-line comments. This manuscript is, in my opinion, quite close to publication-ready.

1. Damage type occurrences (Fig. 7 & 10 metric B) are difficult to interpret given the very different number of leaves (or better yet, leaf area) at each site or of each TCT. The more leaves you look at, the more DT occurrences you will find, which seems to be what you’ve found (e.g., more DTOs at Sf, on TCTs E and F, etc). Could you normalize the number of DTOs by sampling effort? This could be as simple as just dividing the # DTOs by the # of leaves.
2. Additional clarification is needed for the GLM methods and results. What does “/” mean in Table 3: that it was not tested or not significant? If the former (i.e., only some of the predictor variables in this table were used in a given models), why were these particular variables chosen? Did you run additional models that were more inclusive (considered more predictor variables in Table 3), but these did not show significant increases in model predictive power or significance? Species seems like an important predictor metric to include in the herbivory GLMs, as it could be a way to account for chemical defenses and evolutionary histories, but it seems not to have been included in any of the models. If this is the case, I encourage you to redo the models and add species. Similarly, given the importance of Locality in M3 and M3’, why was it not included in M4 and M5?
3. I’m still a little concerned about the differences in preservation between the two floras. While the authors have assured me that you should be able to recognize external foliage feeding equally well at both sites, I am still concerned about differential preservation of leaf mines and galls. How much might that affect differences in DT richness, and in particular that Seifhennersdorf has low DT richess compared to other Oligocene floras from Central Europe? Line 948 -954 are important, but so also would an assessment of the ability to recognize mines and galls on these leaves.
4. Give the writing one more critical review for grammatical errors, awkward sentence structure, and unclear sentences.

Line 311, 586, and Table 3: Both AI or IA are used as abbreviations for area index.

Line 338-348: I recommend moving this section up and incorporating it into the “Leaf manual reconstruction” section, as both are required to get leaf length, with, and area measurements.

Line 350-357: I found your explanation of how leaf phenology was determined to be more clear in the rebuttal letter than the explanation in the manuscript. From the rebuttal letter: “Assessment of the leaf life span of fossil-species (evergreen or deciduous) was based primarily on modern living relatives and, when possible, in combination with mean values of leaf mass per area. For most of the fossil-species mean LMA and deductions from modern living relatives are in accordance.” The text in the manuscript makes it seem that LMA was a greater determining factor, with taxonomic information as secondary.

Line 386: Is semi-deciduous the same as semi-evergreen, which is the notation used in Table S3 and S4?

Line 388-391: Just confirming that herbivory index was only measured for damaged leaves and that no undamaged leaves were included in the analysis. If this is not true, please rewrite to clarify.

Line 406-410: I thought you had a great response to the reviewer comment in your rebuttal letter about why PCA was chosen as the ordination method. Why not include it in the manuscript? This information is especially useful for students.

Line 412: M3-M5? Which damage parameter is which?

Line 561-589: The damage frequencies reported here do not match those in Fig. 7 pie charts A1 and A5. A1 reports 4.78% total damage and external feeding damage (all damage appears to be hole or margin feeding), whereas 4.74% and 4.64% respectively are given in the text. The differences appear to be larger in A5.

Line 613-621: Please also report the number of taxa included within each TCT. I’m curious whether the differences in DTrichness are due to morphological differences or potentially different numbers of species being included, and one might expect increased DTrichness with more plant food sources.

Figure 5A: Should the vector labeled “surface” actually be leaf size?

Figure 7: Panel A6: specify n is leaves, as in other panels. Metric B: Clarify y-axis label. When I read “number of damage types,” I thought it meant how many different DTs are in the floras (i.e., damage type richness). I recommend either “damage type occurrences” or “number of damage type occurrences.” This also caught me in line 567-568.

Figure 10: in the pdf for review, the quality is quite poor. Many of the panels have odd gray boxes and/or lines. See also the comment about Metric B from Figure 7.

---

## Round 0.3 · accepted · Accept

· Academic Editor

Accept

I have assessed this revision myself and find it ready for publication. Thank you for your careful responses to the last round of comments and thorough editing, resulting in a much more engaging and interesting manuscript. Please continue to pay careful attention through the production process to any additional small edits that may be needed.